RESEARCH

# The evolution and mutational robustness of chromatin accessibility in *Drosophila*

Samuel Khodursky[1†] , Eric B. Zheng[1†] , Nicolas Svetec[1] , Sylvia M. Durkin[1,2] , Sigi Benjamin[1] , Alice Gadau[1] , Xia Wu[1] and Li Zhao[1*]

†Samuel Khodursky and Eric B. Zheng contributed equally to this work.

*Correspondence:
lzhao@rockefeller.edu

[1] Laboratory of Evolutionary Genetics and Genomics, The Rockefeller University, New York, NY 10065, USA
[2] Present Address: Department of Integrative Biology and Museum of Vertebrate Zoology, University of California, Berkeley, Berkeley, CA, USA

## Abstract

**Background:** The evolution of genomic regulatory regions plays a critical role in shaping the diversity of life. While this process is primarily sequence-dependent, the enormous complexity of biological systems complicates the understanding of the factors underlying regulation and its evolution. Here, we apply deep neural networks as a tool to investigate the sequence determinants underlying chromatin accessibility in different species and tissues of *Drosophila*.

**Results:** We train hybrid convolution-attention neural networks to accurately predict ATAC-seq peaks using only local DNA sequences as input. We show that our models generalize well across substantially evolutionarily diverged species of insects, implying that the sequence determinants of accessibility are highly conserved. Using our model to examine species-specific gains in accessibility, we find evidence suggesting that these regions may be ancestrally poised for evolution. Using in silico mutagenesis, we show that accessibility can be accurately predicted from short subsequences in each example. However, in silico knock-out of these sequences does not qualitatively impair classification, implying that accessibility is mutationally robust. Subsequently, we show that accessibility is predicted to be robust to large-scale random mutation even in the absence of selection. Conversely, simulations under strong selection demonstrate that accessibility can be extremely malleable despite its robustness. Finally, we identify motifs predictive of accessibility, recovering both novel and previously known motifs.

**Conclusions:** These results demonstrate the conservation of the sequence determinants of accessibility and the general robustness of chromatin accessibility, as well as the power of deep neural networks to explore fundamental questions in regulatory genomics and evolution.

**Keywords:** Deep learning, *Drosophila*, ATAC-seq, Chromatin accessibility, In silico mutagenesis, Robustness, *Drosophila simulans*, *Drosophila yakuba*, *Drosophila melanogaster*

## Background

Organisms can vary drastically, despite sharing substantial identity in the coding sequences of their genes. Much of this variation is likely due to changes in the gene expression as a result of differences in genomic regulatory elements such as enhancers and promoters. Indeed, evidence indicates both that regulatory regions are subject to evolutionary constraints [1, 2] and that mutations in regulatory elements can underlie phenotypic evolution [3–6]. Moreover, in humans, disease- and phenotype-associated variants identified through genome-wide association studies (GWASs) often are enriched in regulatory regions [7, 8], further underscoring the importance of regulatory elements to the origins and evolution of specific phenotypes, including those related to disease susceptibility. However, our understanding of regulatory regions remains remarkably imprecise: for example, while promoters and enhancers are typically thought to be hundreds to thousands of base pairs long [9, 10], the minimum sequence length needed to define a typical regulatory element is unclear [11, 12]. As a result of the imprecision of our understanding, practical applications like determining the impact of a given regulatory mutation in a disease remain out of reach.

Active and potentially active regulatory regions, such as enhancers and promoters, typically have high chromatin accessibility [13, 14], which is readily measured through sequencing assays like ATAC-seq (assay for transposase-accessible chromatin with high-throughput sequencing; Buenrostro et al., 2013 [15]). Chromatin accessibility is a function of the genomic sequence's regulatory potential and transcription factor (TF) binding, but predicting it a priori is difficult, as it and TF binding are entangled: pioneer transcription factors direct chromatin accessibility, and accessibility in turn influences further TF binding [14, 16, 17]. The resulting complexity in the sequence determinants underlying chromatin accessibility thus requires approaches far more sophisticated than the position-weight matrices that can describe TF binding.

Sequence-based machine learning models can be used to decipher the sequence determinants underlying chromatin accessibility [18–25]. Similar models have been used to predict enhancer activity, TF binding profiles, and gene expression from sequence alone [26–30]. Many of these models employ convolutional neural networks, which were initially adapted from image recognition applications and show exceptional promise as sequence-based models for a large variety of genomic prediction tasks. However, few studies have used convolutional neural networks to study the conservation and innovation of accessibility in multiple closely related species.

Due to their inspiration from the mammalian visual processing system, convolutional neural networks are able to consider the position and local sequence context while they learn highly complex nonlinear functions. As an example, in the first layer of a sequence-based convolutional neural network, the trained convolutional filters can behave in a similar fashion to position-weight matrices scanning for motifs [20]; the upper layers in the network then aggregate information across progressively larger contiguous regions of the overall input sequence. The recent addition of self-attention layers allows neural networks to model distant interactions (such as motif-motif interactions) across the entire length of the input sequence without the need for many convolutional layers—in fact, no convolutional layers are necessary in some cases [27, 31]. These advantages and successful prior applications demonstrate that

convolutional neural networks can enable accurate modeling of chromatin accessibility as a function of DNA sequence.

Such accurate computational models could support a variety of in silico experiments to study the sequence-dependent determinants of chromatin accessibility—questions that would be intractable in vivo or even in vitro. Familiar genetic approaches like saturation mutagenesis or knock-ins/knock-outs can be done in seconds, allowing unprecedented scope to study how individual nucleotides, motifs, and sequences contribute to chromatin accessibility. For example, there is considerable interest in identifying factors that contribute to the robustness of transcription regulation to various genetic insults [32]. However, the mutational robustness of regulatory elements has not been systematically investigated at a genome-wide scale in multi-cellular eukaryotes. Here, we collect comparative ATAC-seq data across *Drosophila* and train hybrid convolution-attention neural networks to accurately model chromatin accessibility and, by extension, regulatory elements. We demonstrate that accessibility is broadly conserved across millions of years of evolutionary diversification, likely in large part due to the general mutational robustness of chromatin accessibility in *Drosophila* at a variety of mutational scales.

## Results

### Chromatin accessibility can be accurately predicted from DNA sequence in Drosophila

We generated ATAC-seq libraries from testis and head tissue in *D. melanogaster*, *D. simulans*, and *D. yakuba*. A total of 42.17 gigabases of reads were mapped to the genome, with an average of 31.19 million mapped and properly paired reads per replicate per sample (Additional file 1). Using MACS2 [33], we called peaks indicating accessible chromatin at a false discovery rate (FDR) of 0.01. We identified similar numbers of peaks across all three species (*D. melanogaster*: 26,193; *D. simulans*: 29,656; *D. yakuba*: 28,670). The proportion of peaks in each tissue was similar between *D. melanogaster* and *D. simulans*; in contrast, *D. yakuba* had relatively more peaks in the testis and fewer in the head than in the other two species (Additional file 2: Fig. S1A). The sizes of peaks were broadly similar, with median peak lengths ranging from 386 bp (*D. simulans*) to 443 bp (*D. melanogaster*; Additional file 2: Fig. S1B). Finally, the distribution of peaks across the X chromosome and the autosomes was qualitatively similar across the three species (Additional file 2: Fig. S1C).

To determine if chromatin accessibility could be predicted from sequence alone, we trained multi-task deep neural networks to predict the presence of ATAC-seq peaks in the testis and head given a sequence; this is thus a binary classification problem (Fig. 1A). Training this type of machine-learning algorithm requires a large set of input values ("examples") with known results ("labels"). For binary classification, examples are labeled as either positive or negative. We defined our positive examples as 1-kb one-hot-encoded DNA sequences centered on peaks called by MACS2, while our negative examples consisted of tiled 1-kb one-hot-encoded genomic sequences that did not overlap called peak regions (inaccessible regions; "non-peak" for brevity). Given an example, each distinct neural network ("model") outputs a decimal value between 0 and 1, for both the head and the testis. These model outputs are effectively an accessibility "score" calculated by the model for that sequence and can be considered a predicted probability of accessibility in that tissue. The training process iteratively adjusts internal variables

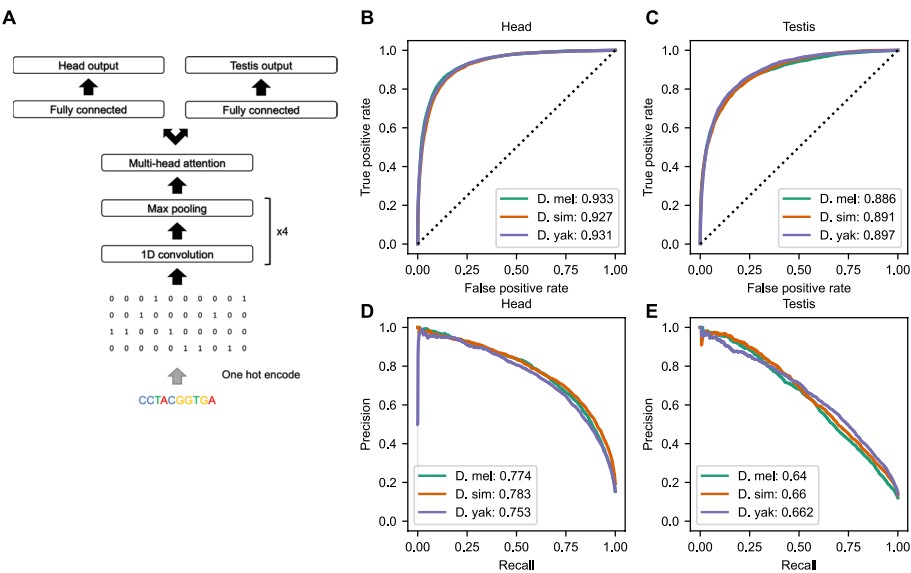

**Fig. 1** Overview of deep learning models and performance. **A** Overview of the model architecture; the same architecture was used for models in all three species. **B** ROC curve showing the classification performance in the head. **C** ROC curve showing the classification performance in the testis. **D**, **E** Precision-recall curves showing the performance in the head and testis, respectively

within the model ("parameters") to minimize errors. We trained one model for each of the three species (three models overall), using examples from chromosome arms 2L, 2R, 3R, X, and 4 from the genome of the respective species. We did not use examples from chromosome arm 3L for training so that we could use them to evaluate model performance. Notably, chromosome arm 3L is syntenic across all three species. Our model architecture consisted of 4 convolutional layers followed by an attention layer and a single fully connected layer (Fig. 1A). We chose hyperparameters such as the convolutional filter size and the number of filters per layer using the tree-structured Parzen estimator (TPE) approach (see the "Methods" section).

Using the area under the receiver operating characteristic curve (AUROC) and the area under the precision-recall curve (AUPR) as classification performance metrics, we found that the trained models could accurately predict chromatin accessibility in both tissues (Fig. 1B–E) (AUROC: head 0.93, testis 0.89–0.90; AUPR: head 0.75–0.78, testis 0.64–0.66). We also trained four other models using *D. melanogaster* data, each leaving one major chromosome arm out for testing. We found that model performance across all models was very similar to the model which used 3L for testing—regardless of which chromosome arm was reserved for model evaluation (Additional file 2: Fig. S2). All downstream analyses were performed using a model trained on all major chromosome arms except 3L and tested on 3L.

Given that regulatory regions generally have higher GC content than non-regulatory regions [34, 35], a major concern would be whether the models are trivially learning the GC content of each example. To test this, we calculated AUROCs and AUPRs based on GC content. We found that GC content was a poor predictor of chromatin accessibility in *Drosophila*, indicating that the models learned far more information than GC content (Additional file 2: Fig. S3) (AUROC: head 0.52–0.58, testis 0.53–0.57; AUPR: head

0.16–0.19, testis 0.13–0.18). Similarly, we wondered if the implicit genomic location for non-peak examples significantly influenced model performance; that is, if genomic sequence varied in some other way with the spatial arrangement of accessible and inaccessible regions (e.g., in euchromatin and heterochromatin), the models might only learn that association. To this end, we only considered non-peak examples whose edges were located at most 500 bp from a peak example and found that model performance was qualitatively unchanged (AUROC: head 0.90–0.91, testis 0.85–0.86; AUPR: head 0.85–0.88, testis 0.76–0.78) (Additional file 2: Fig. S4), suggesting that the models are capable of predicting changes in accessibility within distances less than 1.5 kb along the genome. The increase in AUPR is likely due to the test data becoming less imbalanced when considering only non-peak examples near peaks.

### Chromatin accessibility and its sequence-based determinants are conserved

Using a custom reference-agnostic multi-context approach (see the "Methods" section), we identified the orthologous genomic regions for every peak identified across all three *Drosophila* species. We then compared these regions in the context of every species' genome to determine orthology relationships: we assumed that peaks (or their orthologs) that overlapped in any genomic context were orthologous. In this way, we defined a peak as conserved when its orthologous regions were also accessible, and we defined a peak as species-specific when its orthologs were never accessible. Most peaks (about 75%) are conserved across species; the remainder is divided between species-specific peaks and species-specific losses, with modestly more species-specific peaks in *D. yakuba* and modestly fewer species-specific losses in *D. yakuba* (Fig. 2A). Considering each tissue individually, the relative conservation of peaks is qualitatively similar. Unsurprisingly, peaks observed in both tissues are far more likely to be conserved across all three species (Additional file 2: Fig. S5).

Given that peaks were relatively conserved across species, we wondered if the sequence determinants underlying chromatin accessibility were also conserved. For each of the three models trained in a given species, we examined its performance on data from chromosome arm 3L in the other two species. To compare the three models' relative performance in other species, we took the ratio of each model's AUROC in the two non-native species to the AUROC in its native species. We similarly computed the ratio of AUPR scores for each combination of trained model and species. Since chromosome arm 3L is syntenic in all three species and was excluded from the training data for each model, there is relatively little information leakage resulting from test examples in one species being homologous to training examples in another. When comparing cross-species model performance, the AUROC and AUPR ratios were near 1 in most cases, indicating that the sequence determinants of chromatin accessibility are highly conserved across the three *Drosophila* species (Fig. 2B, C and Additional file 2: Fig. S6).

Because the three species of interest are only separated by several million years of evolution, we wondered if our models could also predict chromatin accessibility in a more distantly related species of *Drosophila*. Thus, we tested our models on ATAC-seq data generated from the heads of *D. suzukii* and found that all three models could accurately predict chromatin accessibility (Fig. 2D). We also tested if our models could predict chromatin accessibility in the very distantly related species *Aedes aegypti* (yellow fever

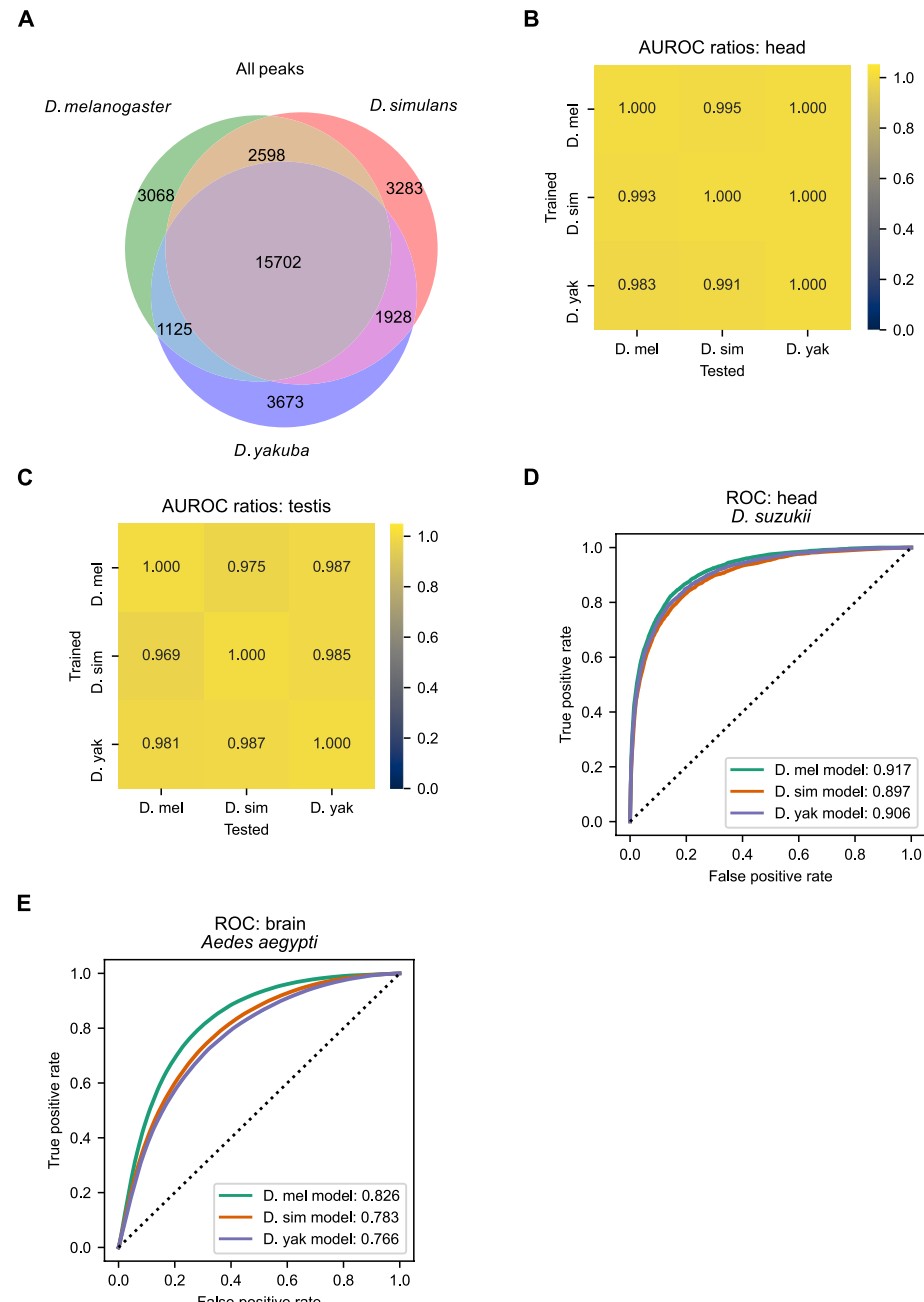

**Fig. 2** The sequence determinants of chromatin accessibility are highly conserved. **A** Venn diagram of the peak conservation across species. We combined the sets of peaks from the head and testis into a single set per species and used a reference-free multi-context approach to identify the orthologous regions for each peak in the other two species (see the "Methods" section). Between two-thirds and three-fourths of peaks identified in each species are conserved. **B**, **C** Each model (trained in a given species) was tested in the other two species. The AUROC of each model tested in a non-native species was divided by the AUROC of the native model. **D**, **E** The models can predict chromatin accessibility in the head of *D. suzukii* and the brain of *Aedes aegypti*

mosquito; last common ancestor with *Drosophila* 150–200 million years ago; [36]) using ATAC-seq data from brain tissue [37]. We found that, despite the large evolutionary divergence and slightly different tissues (heads vs. brains), our models had substantial

predictive power (AUROC: 0.77–0.83; Fig. 2E), further supporting the conclusion that the sequence determinants of chromatin accessibility are highly conserved.

### Species-specific changes in chromatin accessibility may occur at regions ancestrally poised for changes

In addition to the high degree of conservation of chromatin accessibility across species, we also observed regions where chromatin accessibility has likely changed between species (Fig. 2A). To investigate these changes in chromatin accessibility, we closely examined the model behavior across all three models at species-specific peaks (that is, peaks found in one species whose orthologous sequences are inaccessible in both of the other species; see the "Methods" section). Notably, these species-specific peaks are not more divergent in sequence than conserved peaks, suggesting that the phenotypic change in accessibility is not a mere effect of greater genetic change (Additional file 2: Fig. S7). In all combinations of tissues and species, species-specific peak sequences had model outputs from their native model that were significantly lower than the model outputs for all conserved peak sequences, yet species-specific peaks' model outputs remained significantly higher than all non-peak sequences (Fig. 3; one-sided Mann–Whitney $U$-test $p$-values $< 1\mathrm{E} - 10$ for all). We then examined the model outputs for the orthologous inaccessible sequences of species-specific peaks. Surprisingly, when evaluated with the same model as the matched species-specific peaks, the orthologous inaccessible non-peak sequences yielded model outputs significantly higher than all non-peak sequences (one-sided Mann–Whitney $U$-test $p$-values $< 1\mathrm{E} - 10$); interestingly, these model outputs were modestly lower than those for species-specific peaks (*yakuba*-specific peaks in the head and testis: one-sided Mann–Whitney $U$-test $p < 0.001$; *simulans*-specific peaks in the head and testis; *melanogaster*-specific peaks in the testis: $p < 0.05$; *melanogaster*-specific peaks in the head: n.s.). Thus, assuming parsimony, our results suggest that new peaks tend to arise in certain regions that already possess some peak-like properties; that is, these regions may be ancestrally poised for evolutionary change from inaccessible chromatin to open chromatin.

We also examined the species-specific losses for evidence of a similar phenomenon. Since *D. melanogaster* and *D. simulans* are more closely related than *D. yakuba*, the loss of a *D. yakuba* peak in exactly one of the former two species is, by parsimony, species-specific. The loss of accessibility, then, would be a derived trait occurring only in the lineage leading to that species. We do not consider peaks lost only in *D. yakuba*; for these peaks, we cannot by parsimony assume the ancestral state at the last common ancestor of all three species. As with species-specific gains of accessibility, we likewise found that species-specific inaccessible sequences had model outputs significantly higher than those of all non-peak sequences (Additional file 2: Fig. S8; *melanogaster*-specific losses of accessibility in head and testis; *simulans*-specific loss of accessibility in head: one-sided Mann–Whitney $U$-test $p$-values $< 1\mathrm{E} - 10$; *simulans*-specific loss of accessibility in testis: $p < 0.001$). Similarly, the orthologous accessible sequences had model outputs significantly lower than all conserved peak sequences (orthologous peaks of *melanogaster*-specific losses in head and testis; orthologous peaks of *simulans*-specific losses in testis: $p < 1\mathrm{E} - 10$; orthologous peaks of *simulans*-specific losses in head: $p < 0.001$). Moreover, in *D. melanogaster*- and *D. simulans*-specific losses in the head, model outputs from

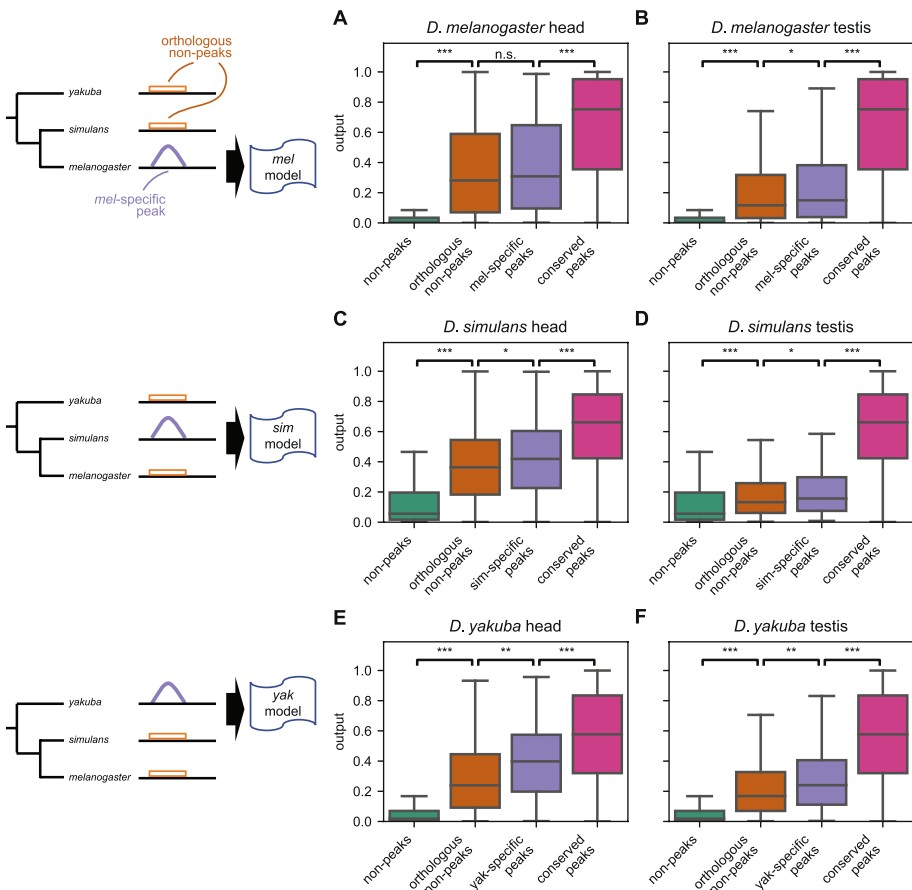

**Fig. 3** Species-specific peaks may be poised for evolution. **A**, **B** Boxplots of raw model outputs in the head (**A**) and testis (**B**) from the model trained on *D. melanogaster* data. *D. melanogaster*-specific peaks on 3L (purple) have significantly lower model outputs than conserved peaks overall (magenta); their orthologous non-peak sequences (orange) from *D. simulans* and *D. yakuba* have significantly higher model outputs than non-peaks overall (green). The species-specific peaks have modestly higher model outputs than their orthologous non-peaks. The phylogeny is not to scale. One-sided Mann–Whitney *U* tests; ***: $p < 1E - 10$; **: $p < 0.001$; *: $p < 0.05$. **C**, **D** As **A** and **B**, except using the *D. simulans* model, the *D. simulans* 3L peak (magenta) and non-peak examples (green), *D. simulans*-specific peaks (blue), and their orthologous non-peak sequences from *D. melanogaster* and *D. yakuba* (orange). **E**, **F** As **A** and **B**, except using the *D. yakuba* model, the *D. yakuba* 3L peak (magenta) and non-peak examples (green), *D. yakuba*-specific peaks (blue) and their orthologous non-peak sequences from *D. melanogaster* and *D. simulans* (orange)

species-specific inaccessible sequences were modestly lower than that of their accessible orthologs (*simulans*-specific inaccessible sequences vs. orthologous peaks: $p < 0.05$; *melanogaster*-specific losses: n.s.), even though the accessible peak sequences were now from a different species compared to the model's training data. This suggests that the difference in outputs is not due to orthologous sequences having systematically different model outputs, e.g., as a result of a mismatch between the orthologous genome sequences and the genomes on which the models were trained. Thus, when chromatin accessibility is lost, the ancestral peak may have been poised for inaccessibility. Together, these results suggest that changes in chromatin accessibility—both gains and losses—may occur in regions that are ancestrally poised for change.

False-negative identifications of peaks when comparing species could be a confounder: to make this conclusion of ancestral evolutionary poise, we must assume that inaccessible regions in another species are truly inaccessible. To mitigate this, we first examined only the *D. melanogaster*-specific peaks with the weakest chromatin accessibility in the other two species. We defined high-confidence species-specific peaks by selecting peaks whose coverage in the raw ATAC-seq data in the orthologous non-peak regions in *D. simulans* and *D. yakuba* was less than the first percentile of peaks called in both those species (Additional file 2: Fig. S9). Once again, our results were qualitatively similar: the orthologous non-peak sequences from *D. simulans* and *D. yakuba* retained higher model outputs than non-peak sequences overall, and the *D. melanogaster*-specific peak sequences retained lower model outputs than *D. melanogaster* peak sequences overall. Secondly, we bypassed the peak-calling process outright by directly using ATAC-seq coverage (*i.e.*, Tn5 insertion density). We normalized coverage data and model outputs by calculating the percentile ranks of each, and we examined the relative change in percentile rank of model outputs along with the relative change in percentile rank of ATAC-seq coverage between *melanogaster*-specific peaks and their orthologous non-peak sequences. We expected that true-positive peaks would have the largest percentile rank change in ATAC-seq coverage between *D. melanogaster* and the other two species. Looking specifically at those peaks with the largest change in coverage percentile, we did not observe an increase in change in predicted model output percentiles (Additional file 2: Fig. S10A). Moreover, those orthologous non-peak sequences with the largest change in coverage percentiles still have model outputs between peaks and non-peaks (Additional file 2: Fig. S10B). Thus, it is unlikely that species-specific peaks are a spurious result of missed peak calls in the other species.

### Saturation mutagenesis reveals that inaccessible regions show evidence of evolutionary constraint acting to keep chromatin inaccessible

To aid in our interpretation of the *D. melanogaster* model, we performed saturation in silico mutagenesis on all test examples (from chromosome arm 3L). For each test example, we calculated model outputs for every possible single base-pair substitution and compared them with the reference model outputs. The mutational effects had a heavy-tailed, approximately log-normal distribution (Fig. 4A). This implies that while most mutations have little effect, a small proportion of sites with substantial effects may largely determine chromatin accessibility in any given example. In contrast, under a non-heavy-tailed distribution, such as an exponential distribution, one would expect a greater proportion of sites with more moderate effects to be largely responsible for chromatin accessibility. We then calculated the importance scores for each base pair, where the importance score is the difference in log space between the odds of the model output with the reference base compared to the average odds of the model output across all 4 possible bases (see the "Methods" section). Since the model output can be interpreted as the probability of a peak given that sequence, the importance score of a given site thus reflects the relative contribution of that site to the probability of a peak overall. Positive importance scores indicate that the reference base is predicted to make the chromatin more accessible than a randomly chosen base. Likewise, a negative score indicates that the reference base is predicted to make the chromatin less accessible than a randomly

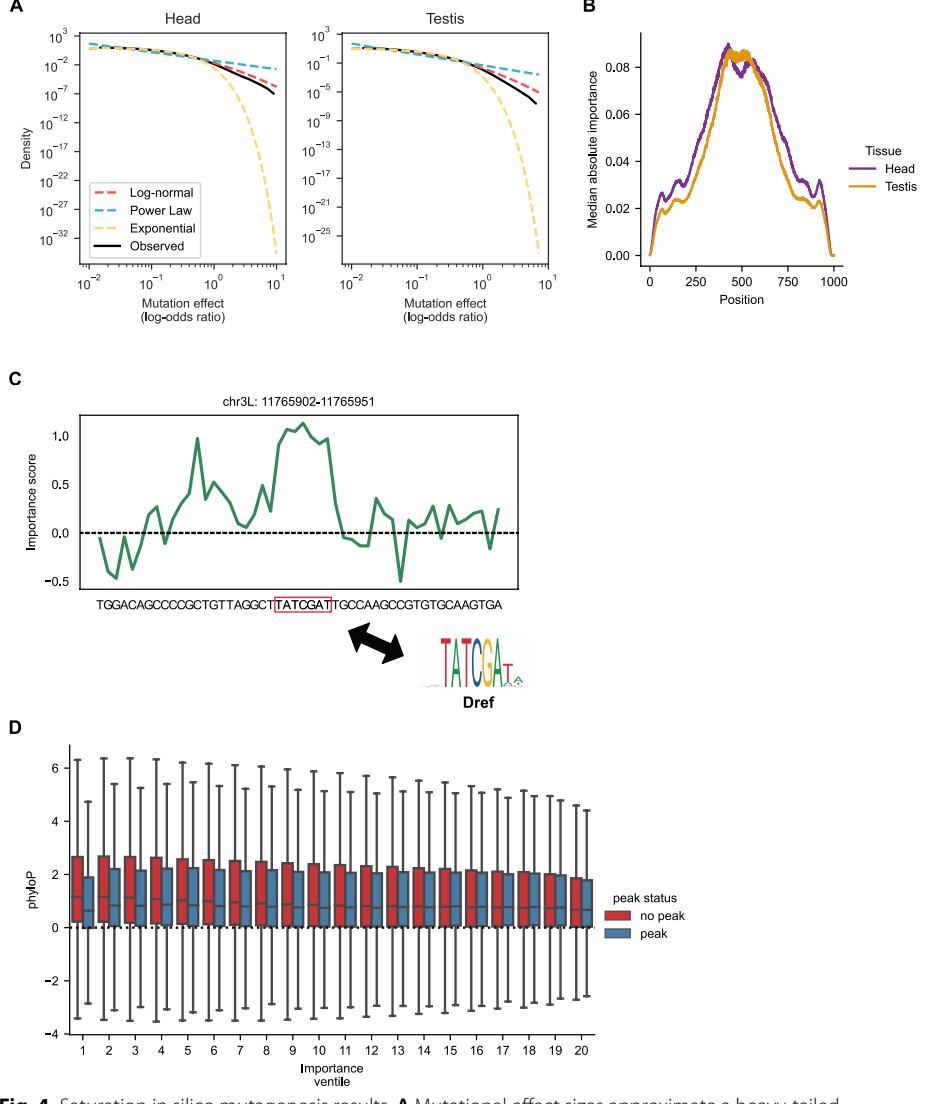

**Fig. 4** Saturation in silico mutagenesis results. **A** Mutational effect sizes approximate a heavy-tailed distribution (log-normal). **B** The median absolute importance score binned by position. **C** An example of importance scores for a peak region on chromosome 3L. The maximum importance score for the entire example overlaps a *Dref* motif. **D** There is a significant negative correlation between importance scores and phyloP (Spearman's $r = -.085$, $p < 1e - 100$) in inaccessible regions, indicating that purifying selection is acting to keep inaccessible regions closed. Only the central 200 bp of each example was considered. Outliers have been omitted for clarity

chosen base. By plotting median importance scores against positions within examples, we found that essentially all positions within an example contributed to the model's predictions. However, the central 200 bp were the most important in both tissues (Fig. 4B). An example of importance scores for a peak region is shown in Fig. 4C. The highest importance scores for the entire example overlap a binding motif for DNA replication-related element factor (Dref), a known chromatin organization regulator [38].

Prior work used deep learning models to link chromatin accessibility to evolutionary constraints in human cell lines [20]. Thus, we hypothesized that we could use our model to observe signatures of evolutionary constraints acting on chromatin accessibility in

*Drosophila*. By adding importance scores between the two tissues, we observed a small but significant negative correlation between importance scores and phyloP scores in inaccessible regions (Spearman's correlation $r = -0.085$, $p < 1e-100$) (Fig. 4D). Since phyloP scores are a base pair-level measure of sequence conservation—more positive values indicate increased conservation while more negative values indicate accelerated evolution—a negative correlation between phyloP and importance scores in inaccessible regions implies negative selection against mutations which make inaccessible regions more accessible. Similarly, negative selection acting to remove mutations which make accessible regions inaccessible would result in a positive correlation between phyloP and importance score. Surprisingly, we also observed a small but significant negative correlation between importance scores and phyloP scores in peak regions (Spearman's correlation $r = -0.016$, $p < 1e-21$) (Fig. 4D). This observation could be caused by multiple factors. For example, we observed that importance scores tended to be of a larger magnitude in non-peak examples than in peak examples (Additional file 2: Fig. S11). Larger importance scores imply larger mutational effects. Therefore, it is possible that a lack of positive correlation between phyloP and importance scores could be attributed to the sensitivity of our approach in detecting weak evolutionary patterns. Additionally, the weak negative correlation could be biologically meaningful, suggesting that peak regions have experienced slightly accelerated evolution.

### In silico knock-in/knock-out mutagenesis reveals that chromatin accessibility can be predicted from very short contiguous sequences

Since saturation in silico mutagenesis revealed that, on average, the central 200 bp of each example were most critical, we wondered whether chromatin accessibility could be predicted using short sequences from each example. To determine short contiguous subsequences predictive of chromatin accessibility, we first found the most important subsequence in each example by identifying the subsequence with the highest moving average of importance scores from our saturation in silico mutagenesis that would change predicted identity. Specifically, for predicted non-peak examples, we considered the moving average window position that most increased model output; conversely, for predicted peak examples, we considered the window position that most decreased the model output. Notably, we inferred accessibility status for each example using only the model outputs, instead of the labels, to avoid information from the labels influencing, and likely inflating, the importance of the identified sequences. We inferred the accessibility status of each example by binarizing model outputs according to a threshold. We determined this threshold for each tissue by maximizing the product of the sensitivity (true positive rate) and specificity (1 minus the false-positive rate) in the validation set. The most important subsequences were then inserted into the center of a constant set of 10 arbitrary non-peak examples that served as an in silico background for insertion. The discriminative power of the subsequences was measured using the AUROCs and AUPRs using the original example labels (Fig. 5A, Additional file 2: Fig. S12A). Surprisingly, we found that even 5-bp and 10-bp subsequences had substantial discriminative power.

We then wondered if knock-out of the most important subsequences in each example (in their native context) substantially impaired model performance. Knock-out was performed by replacing the original subsequences with random sequences. We found

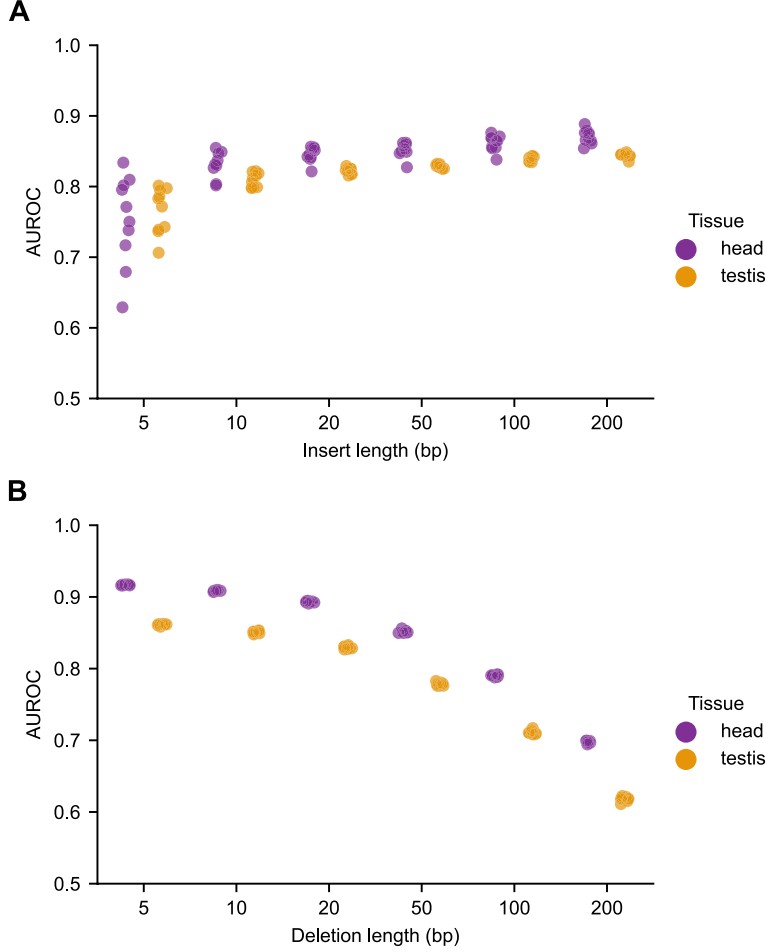

**Fig. 5** Chromatin accessibility can be accurately predicted from short contiguous subsequences. **A** Short subsequences identified from in silico mutagenesis are predictive of chromatin accessibility. For each example, the short subsequence corresponding to the window with the largest moving average importance was inserted into a constant genomic background. Ten sequences with low baseline model outputs were randomly selected, and AUROCs were calculated for model performance. **B** Deletion of windows with the highest importance in each example did not substantially diminish model performance for small windows. Deletions were performed by substituting the native subsequence with a random sequence, and AUROCs were calculated for model performance

that model performance only gradually decreased with increasing length of ablated subsequence (Fig. 5B, Additional file 2: Fig. S12B). Knock-out of the 5 or 10 most important base pairs in each example minimally impaired classification accuracy. These in silico knock-in and knock-out experiments imply that the chromatin accessibility is mutationally robust: deletion of the most important (and predictive) short subsequence in each example did not significantly impair classification. It also suggests the possibility that the sequence underlying chromatin accessibility is somewhat redundant.

## Chromatin accessibility is predicted to be robust to large-scale random mutation

To further examine the mutational robustness of chromatin accessibility, we performed large-scale random mutagenesis on the test examples using the *D.*

*melanogaster* model. We randomly made 50, 100, 200, or 500 substitutions in each example and observed the model outputs. For each example, we performed 1000 mutational experiments to obtain model output distributions at each level of mutagenesis. As a quantitative measure of robustness, we calculated the fraction of experiments in which each example was predicted to change state—from peak to inaccessible and vice versa. The original (model-predicted) state of each example was determined by binarizing model outputs based on a threshold chosen using the validation set, as in the analysis from Fig. 5. Following mutation, if the model output for a given example crossed the threshold, it was considered a state change. We found that even at 20% sequence divergence, corresponding to 200 mutations without back-mutation, most peak (Fig. 6) and non-peak (Additional file 2: Fig. S13) examples in both tissues were predicted to retain their peak state more than 50% of the time. A concern with using machine learning in this context is that our model may be brittle to large-scale mutations, since the resulting mutated sequences may be so different from the training data as to be not properly accounted for by the model. However, a sequence divergence of 20% is about the average divergence within peaks between *D. melanogaster* and *D. yakuba* (Additional file 2: Fig. S7), and the model predictions remain relatively stable and dramatically different from model predictions for random sequences (Fig. 6, Additional file 2: Fig. S13). These results suggest that the model is not brittle with respect to large-scale sequence mutation at these magnitudes. Overall, the findings indicate that chromatin accessibility is robust to large-scale random mutation. It seems plausible that this robustness is a product of the heavy-tailed distribution of mutational effects (Fig. 4A): most mutations have minimal effects, with only a few possible mutations having large effects.

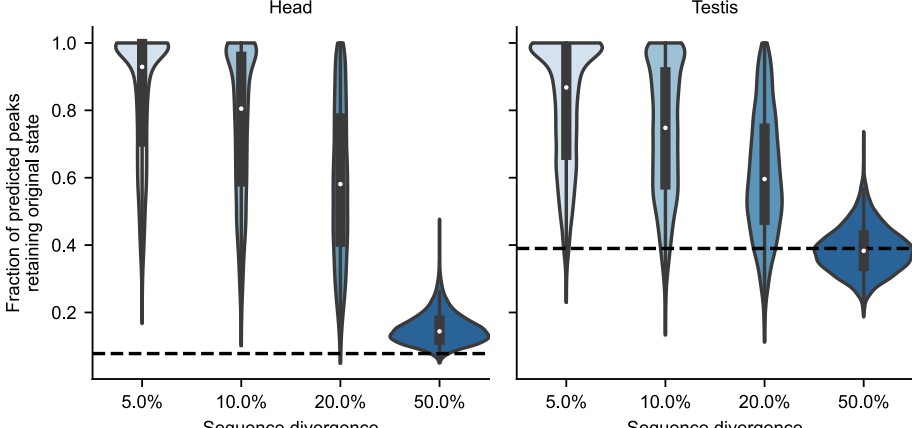

**Fig. 6** Chromatin accessibility is predicted to be robust to large-scale mutation. Most peaks are predicted to remain as peaks more than 50% of the time after 20% of the sequence has been randomly mutated. At each mutational level, 1000 experiments were run for each example. For each example, the fraction of experiments in which the peak was predicted to remain a peak was calculated. The violin plots show the distributions of these fractions across all peak examples. The dotted line represents the fraction of experiments in which a completely random sequence was classified as a peak

**Strong selection in silico demonstrates the malleability of chromatin accessibility and the evolutionary constraints imposed by multiple tissues**

While the previous mutagenesis experiments investigated the ability of chromatin accessibility to resist change in the presence of mutation, we also wondered if chromatin accessibility could rapidly change under positive selection. We evolved examples in silico under a strong-selection weak-mutation (SSWM) regime. Under SSWM, only the fittest individual survives to reproduce, and the mutation rate is low enough that each individual only acquires a single mutation from the previous generation. In practice, for examples where selection acts to increase the accessibility of a sequence, all possible mutations a single mutational step away from the extant sequence are made, and the model outputs are obtained. The mutation which produces the largest increase in predicted accessibility is retained as the starting sequence for the next generation. For 1000 randomly selected examples which were non-peaks in both tissues, selection for accessibility in head tissue resulted in rapid accessibility gains over a few mutational steps in both head and testis tissue, even though selection was not applied to testis tissue (Fig. 7A). This result simultaneously indicates that chromatin accessibility is highly malleable under strong selection and that selection in one tissue will "drag" the accessibility phenotype in the second tissue in the absence of selection in that tissue. Selection for accessibility in both tissues simultaneously similarly led to large gains in accessibility over a few generations (Fig. 7B).

Next, we investigated the constraint imposed by differing directions of selection in the two tissue mutations. Now, we maximized the difference in accessibility between head and testis tissues in each generation (see the "Methods" section). We found that accessibility in the head increased while accessibility in the testis remained low. However, the number of generations required for accessibility to arise in head tissue increased compared to SSWM without selection in testis (an unconstrained condition), demonstrating that tissue-specific selection patterns can slow adaptation (Fig. 7C). Even though in both conditions (selection only in head and tissue-specific selection), accessibility in head tissue increased, the mutational paths taken by identical examples in the two conditions differed substantially (Fig. 7D). For the majority of examples tested, fewer than 10% of the adaptive mutations made in the unconstrained condition were also made in the constrained condition. The existence of multiple, essentially mutually exclusive, mutational paths from inaccessibility to accessibility further shows the lability of chromatin accessibility and can help explain the prevalence of tissue-specific peaks.

We then performed SSWM experiments on accessible sequences (Additional file 2: Fig. S14). Like our SSWM experiments on inaccessible regions, chromatin accessibility was predicted to be lost after only a few mutations in previously accessible sequences. Selection in head tissue rapidly reduced accessibility in testis tissue (Additional file 2: Fig. S14A), and selection in both tissues led to a similar result (Additional file 2: Fig. S14B). A difference between our two sets of SSWM experiments (on initially accessible vs. inaccessible sequences) was that opposing selection on peaks (selection for inaccessibility in head and accessibility in testis) resulted in slower and less efficient emergence of tissue-specific peaks (testis-specific peaks in this case) (Additional file 2: Fig. S14C). Additionally, there was a larger overlap in mutations between selection in head only vs opposing selection in head and testis (Additional file 2: Fig. S14D). This indicates that

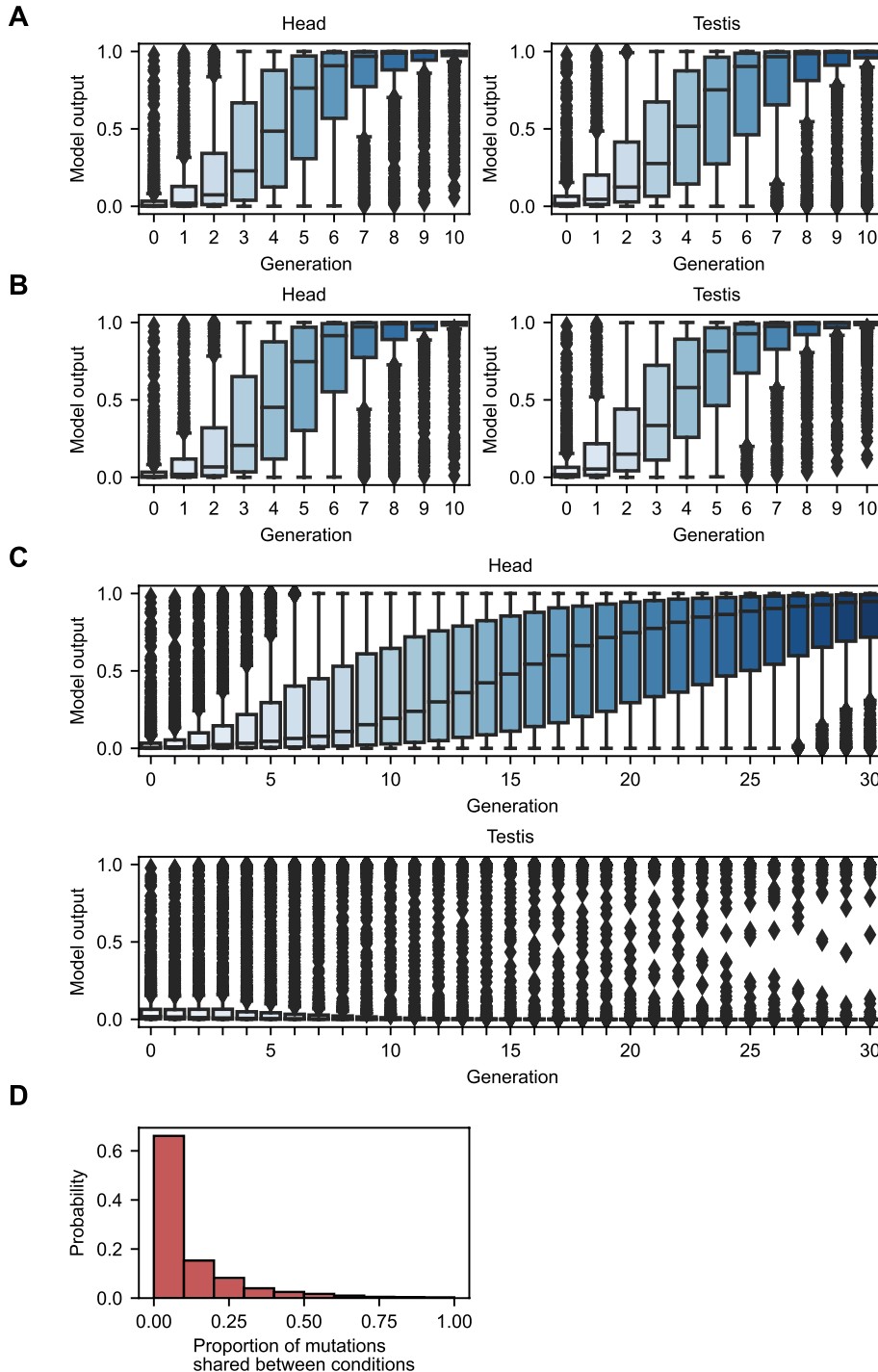

**Fig. 7** Strong-selection weak-mutation experiments on inaccessible sequences. **A** Strong selection for accessibility in head tissue on 1000 randomly selected non-peak sequences (non-peaks in both tissues) results in rapidly increasing accessibility in both head and testis tissues. **B** Using the same examples as in **A**, strong selection for accessibility in both tissues results in rapidly increasing accessibility in both tissues. **C** Using the same examples as in **A**, strong selection for accessibility in the head and inaccessibility in the testis results in head-specific accessibility; however, adaptation is slowed compared to **A**. **D** The proportion of mutations made to examples in **A** also made in **C**, revealing distinct mutational paths to accessibility in head tissue

tissue-specific selection can, in some cases, substantially slow the rate and ability of chromatin-specific accessibility to arise. The asymmetry between the results in Fig. 7 and Additional file 2: Fig. S14 is reflected in the abundance of head-specific peaks compared to testis-specific peaks in the empirical data in *D. melanogaster*.

### Identification of motifs predictive of chromatin accessibility

Finally, we set out to identify transcription factor binding motifs that were predictive of chromatin accessibility in each tissue. We used our saturation in silico mutagenesis scores as inputs into TF-MoDISco—a clustering-based approach which allows one to recover general motifs which tend to influence model predictions [39]. By using TF-MoDISco in the multi-task setting, and using predictive motifs as queries in the JASPAR Core Insects database [40], we identified motifs corresponding to 15 distinct transcription factors (FDR < 0.05) (Additional file 3). Among these, we identified motifs for the GAGA factor (*GAF*/*Trl*), caudal (*cad*), and tramtrack (*ttk*) (Fig. 8A–C). The *GAF* binding motif was predictive of increased accessibility in both tissues (Fig. 8A); *GAF* is a pioneer factor in *Drosophila* that has been shown to be critical for early embryonic development [41–43]. Mechanistically, *GAF* has been shown to directly interact with the PBAP chromatin remodeling complex to open chromatin and recruit RNA polymerase

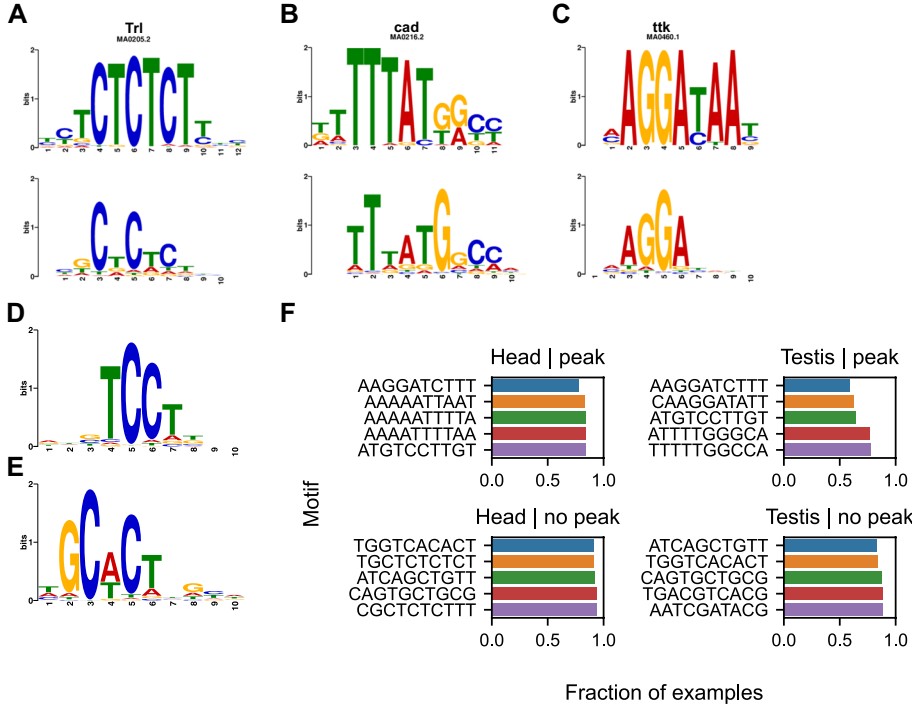

**Fig. 8** Motifs predictive of chromatin accessibility. **A–C** Motifs identified by TF-MoDISco are shown at the bottom of each panel while known motifs from the JASPAR insect collection are shown at the top of each panel. **A** A motif corresponding to Trl was predictive of open chromatin in both tissues. **B** The *cad* motif was predictive of closed chromatin in both tissues. **C** The *ttk* motif was predictive of closed chromatin in the testis. **D–E** Motifs predictive of inaccessibility (**D**) and accessibility (**E**) that do not significantly align with motifs in the JASPAR database. **F** The effect of inserting TF-MoDISco identified consensus motifs predictive of accessibility and inaccessibility into the center of non-peak and peak examples, respectively. The five motifs with the largest median impact on model output in each condition are shown

II to promoters [44]. Additionally, *cad* and *ttk* motifs were predictive of decreased accessibility in both tissues and in the testis, respectively (Fig. 8B, C). *Ttk* has been experimentally shown to interact with *GAF* and to repress *GAF*-mediated activity [45]. These results show that our model discovered biologically meaningful patterns and highlight the importance of both activating and repressive motifs in determining chromatin accessibility. In addition to known motifs, we identified 90 motifs that were predictive of accessibility but could not be aligned significantly to motifs in either the JASPAR Core or JASPAR Core Insects database (FDR > 0.05) (Additional file 3). Two such motifs are shown in Fig. 8D, E.

To further investigate the robustness of the sequence determinants of chromatin accessibility to large-scale mutation, we sought to determine if a single motif predictive of chromatin accessibility could substantially change the predicted accessibility of a sequence. We inserted consensus motifs identified with TF-MoDISco into the center of test examples and measured the change in model output. Specifically, we inserted sequences predictive of increased accessibility into non-peak examples, and we inserted sequences predictive of decreased accessibility into peak examples. Predicted changes in chromatin state (peak to inaccessible and vice versa) for each example were calculated as in Fig. 6. Overall, the insertion of predictive motifs changed the predicted chromatin state in fewer than 41% of peaks for all testis-derived motifs and in fewer than 23% of peaks for all head-derived motifs (Fig. 8F). Non-peaks changed state at an even lower rate. These results, along with the results in Figs. 5 and 6, imply that the sequence determinants of accessibility are largely robust to a variety of genetic perturbations.

## Discussion

In this study, we trained deep learning models to accurately classify chromatin accessibility in *Drosophila* using only local DNA sequences as input. We found that GC content and the distance of non-peak examples from peak examples had little influence on model performance, indicating that the models had learned non-confounded sequence patterns that can non-trivially predict accessibility. Another possible concern is that the model is simply discriminating between constitutive heterochromatin and other chromatin states. Recent sources have suggested that up to 30% of the *D. melanogaster* genome could be constitutive heterochromatin [46]. However, much of that heterochromatin is not included in the Release 6 version of the reference genome (regions not included in the reference genome are not relevant to our analysis, as they are not included in either training, validation, or testing), which is the reference that our work is based on. Marsano *et al.* estimate that ~ 16–17 Mb of the R6 reference is constitutive heterochromatin, excluding the Y chromosome (which we exclude in our own analysis) [46]. In contrast, our dataset (training + validation + testing) consists of over 100 Mb of nonpeak examples for each tissue. Therefore, it is implausible that our model simply discriminates between constitutive heterochromatin and all other chromatin states.

By training models in one species and testing them in another, we showed that the sequence determinants of chromatin accessibility are highly conserved between *D. melanogaster*, *D. yakuba*, and *D. simulans*, and even as far as *Aedes*. This result was consistent with previous findings suggesting that sequence patterns predictive of enhancers are conserved across mammalian species [26].

In general, chromatin accessibility is conserved evolutionarily, with most peaks retained across all three species. We found evidence that changes in chromatin accessibility between species may occur at sites ancestrally poised for change. Model outputs for species-specific peaks were significantly lower than for all conserved peaks; outputs for sites of species-specific losses of accessibility were significantly greater than all inaccessible regions. Moreover, a similar trend was true for the orthologous regions: the inaccessible orthologs of species-specific peaks had greater model outputs than all inaccessible regions, and the orthologous peaks of species-specific losses had lower outputs than conserved peaks. As a result, the apparent change in phenotype appears to be accompanied by a surprisingly small inferred change according to the trained models, and the orthologs appear to have a surprisingly large inferred difference in sequence-specific regulation of accessibility. Notably, the genetic divergence between species at regions with changes in chromatin accessibility similar to those regions with conserved accessibility, reinforcing our intuition that specific sequence-dependent factors govern accessibility. While unlikely, it is possible that orthologous non-peak regions are in fact "weak" peak regions that were incorrectly missed by MACS2; if this were true, it would imply that the sequence-dependent factors that govern accessibility are so extraordinarily conserved across species that our trained models are superior to MACS2 at peak-calling. Instead, we propose that locations with changes in chromatin accessibility are evolutionarily poised to gain or lose chromatin accessibility. These sites could serve as evolutionary hotspots for regulatory evolution. Additionally, the lower model outputs for species-specific peaks (when compared to conserved peaks) could indicate that *trans*-regulatory or more distal sequence factors contribute disproportionately to the evolution of novel peaks—this should be investigated in depth in future studies.

To gain further insight into the sequence determinants of chromatin accessibility, we performed several different types of in silico mutagenesis: saturation mutagenesis, knock-in/knock-out mutagenesis, large-scale random mutagenesis, and in silico evolution under a strong-selection weak-mutation (SSWM) regime. Using saturation mutagenesis, we found evidence that there is negative selection acting to maintain the inaccessibility of inaccessible regions. This selection may act to hinder the evolution of novel accessible regions and help explain our observation of large-scale peak conservation across the three species. These results are also consistent with previous findings showing evidence of selection against spurious TF binding sites in bacteria and archaea [47]. Knock-in/knock-out mutagenesis revealed that certain short subsequences in each example on the scale of 5 bp were predictive of chromatin accessibility. These results imply that regulatory regions such as enhancers and promoters can be defined by sequences on the scale of a single TF binding site. A consequence of this is that regulatory regions may frequently arise by simple chance throughout the genome, necessitating pervasive negative selection to keep aberrant transcription at bay. Indeed, our results from saturation mutagenesis found signatures of this selection. Finally, large-scale mutagenesis at the scale of species divergence suggests that even in the absence of any selection whatsoever, the chromatin accessibility state would be retained in most cases. Thus, in total, it appears as though chromatin accessibility is maintained through a set of mutationally robust and possibly redundant sequence determinants; this is consistent with prior findings with respect to robust regulatory elements [48]. Knock-out of

the most important subsequences, insertion of predictive motifs, and large-scale random mutagenesis did little to qualitatively change model outputs. Negative selection and chromatin accessibility robustness may act together to preserve the overall landscape of chromatin accessibility and in turn transcription. In silico evolution under an SSWM regime revealed that despite the mutational robustness of chromatin accessibility, and consistent with our prior analyses, peaks can arise de novo from inaccessible sequences in only a handful of mutations. However, our SSWM results also demonstrate how tissue-specific constraints on chromatin accessibility can slow adaptation and increase the length of the mutational path required to achieve tissue-specific accessibility. Cell- and tissue-specific constraints on chromatin accessibility thus plausibly substantially slow the adaptation of new peaks in *Drosophila*.

Finally, we identified transcription factor binding motifs predictive of chromatin accessibility in both tissues, consistent with the motifs' previously known functions. We found motifs that were associated with increased and decreased chromatin accessibility; notably, we identified a motif corresponding to the pioneer factor *GAF* as predictive of open chromatin in both tissues. Although *GAF* has previously been shown to directly open chromatin and regulate transcription in embryos and embryo-derived cell lines [42–44], our results suggest that *GAF* may also establish regions of open chromatin in adult tissues. Overall, these results serve two purposes: they show that our model learned genuine biological patterns, and they emphasize the importance of activating and repressing motifs in shaping the chromatin accessibility landscape.

Combining our observations about robustness and ancestral poise suggests a potential mechanism for the evolution of chromatin accessibility. In general, the state of chromatin accessibility appears to have substantial robustness by default, as our models suggest that the vast majority of individual changes and even the accumulation of many individual mutations have minimal effect. In a recent similar study on promoters [49], individual changes similarly tended to have minimal effect, but aggregates of many changes tended towards randomness. Thus, conservation of chromatin accessibility appears common due to robustness, implying that genetic change at regulatory sites can be highly permissive and that evolution at such sites can generally occur under neutrality. Considering changes in chromatin accessibility, we note that neutral evolution would generate substantial genetic variation over evolutionary time and is consistent with our proposition that changes in chromatin accessibility occur at sites of ancestral evolutionary poise. Neutral variation—enabled by robustness—may occasionally, gradually result in an evolutionarily poised state where further evolution can occur. Nevertheless, mutational options with large effect sizes exist and allow for rapid adaptation under strong selective regimes. The observed gradual evolution of chromatin accessibility, however, suggests either that strong directional selection on chromatin accessibility is not particularly common or that tissue-specific constraints substantially inhibit the directional evolution of accessibility. Our study thus showcases the utility of sequence-based deep learning models in generating evolutionary and mechanistic insights into regulatory genomics.

Other details on the nature of ancestral evolutionary poise and accessibility changes remain to be investigated. A model such as ours could be used to investigate the population genetics of chromatin accessibility using data from multiple strains and tissues [50].

Additionally, future studies could consider chromatin state at a finer resolution like the ENCODE, modENCODE, and ChromHMM studies [51–54]. A deep learning model capable of distinguishing between more nuanced chromatin states would allow for additional evolutionary and biological insights.

## Conclusions

We have trained hybrid convolution-attention neural networks using ATAC-seq and DNA sequences as inputs, achieving accurate predictions of ATAC-seq peaks. Our findings indicate that models generated from one *Drosophila* species can be effectively applied to predict open chromatin in another species, maintaining substantial accuracy even when applied to distantly related species such as *D. suzukii* or *Ae. aegypti*. Furthermore, our analyses reveal that species-specific peaks and their non-peak homologs in outgroup species display similar predictive scores, implying a tendency for certain sequences to be predisposed as peaks. This also indicates a constraint on the potential to be open for closed chromatin regions. Moreover, in silico mutagenesis experiments reinforce that the regulatory foundation governing open chromatin conformation is conserved, robust, and malleable under strong selection. Our innovative approach will facilitate an in-depth investigation into the evolution and origins of open chromatin conformation, potentially unveiling novel open chromatin peaks in insect species and beyond.

## Methods

### ATAC-seq library preparation

We performed ATAC-seq using about 2-day-old female heads and male testes from *D. melanogaster* (Bloomington #2057; Drosophila 12 Genomes Consortium et al., 2007 [55]), *D. simulans* (w501; Drosophila 12 Genomes Consortium et al., 2007 [55]), and *D. yakuba* (Tai18E2; Drosophila 12 Genomes Consortium et al., 2007 [55]) and 2-day old female heads from *D. suzukii* (WT3; Durkin et al., 2021 [56]; Chiu et al., 2013 [57]). For each experiment, 25 newly emerged flies per replicate were collected and transferred to new vials; there were three replicates per sample. Two days later, we dissected heads or testes in cold PBS. Sample preparation was performed as described previously [58]. Libraries were constructed using the same primers as [59] and sequenced on a 75-bp paired-end Hiseq X platform.

### ATAC-seq data processing

Raw reads were trimmed using Trimmomatic [60]. Reads were aligned to their respective genomes using Bowtie2 using the following options: "--very-sensitive-local -k 10" [61]. For *D. melanogaster*, the "dmel-all-chromosome-r6.41.fasta" genome was used, for *D. simulans* the "dsim-all-chromosome-r2.02.fasta" genome was used, and for *D. yakuba* the "dyak-all-chromosome-r1.05.fasta" genome was used. All genomes were obtained from FlyBase. Peaks were then called using MACS2 at an FDR < 0.01 with the following options: "-f BAMPE -g dm -q 0.01 --keep-dup all" [33]. The three replicates from each species/tissue were processed together.

### Input data for CNN models

The narrowPeak files generated by MACS2 were processed to generate input data for the CNN models. First, overlapping peaks across the two tissues were merged into a single peak. Then, the midpoint of each peak was taken, and the 500-bp flanking sequences (on each side) were extracted. These sequences were one-hot encoded and were taken to be the positive (peak) examples for the classifier. To generate negative (non-peak) examples, 1-kb tiles spanning the entire genome were generated. Tiles overlapping peaks were thrown out, while the remaining sequences were one-hot encoded and taken to be negative examples. Positive examples were given a label of "1," and negative examples were given a label of "0." The input sequences for both tissues were identical; however, the labels were allowed to be tissue-specific. This allowed us to frame the classification problem as a multi-task problem. Examples from chromosomes 2R, 2L, 3R, 4, and X were used for training while examples from chromosome 3L were used for model testing. We chose 3L for testing, as we wished to have an autosomal chromosome that was syntenic across all three species; a pericentric inversion in *D. yakuba* on chromosome 2 precluded the use of 2L and 2R, and we selected 3L over 3R to preserve more examples for training. We later also tested the performance of the models on all other chromosomes. A random 10% subset of the training data was used as a validation set to determine the model hyperparameters and implement early stopping during training.

### CNN model architecture and training

The CNN models were implemented in Python using the PyTorch library [62]. The models consisted of 4 shared (between the tissues) convolutional layers feeding into a shared multi-head attention (MHA) layer and into a pair of tissue-specific fully connected layers. Prior to input into the MHA layer, data was given a positional encoding as described in Vaswani et al. [31]. Each convolutional layer consisted of convolution followed by batch normalization, a rectified linear unit (ReLU), and max pooling. The fully connected layers consisted of a linear transformation, dropout, ReLU, linear transformation, and sigmoid. The models were trained using a binary cross-entropy (BCE) loss function and stochastic gradient descent (SGD). The total loss function for each model was taken to be the sum of the BCE losses in each tissue.

   The convolutional filter size, max pool size, dropout (for both fully connected and MHA layers), and momentum (for SGD) were chosen using the tree-structured Parzen estimator approach as implemented in the Optuna library [63]. One hundred Optuna trials were run, and the set of hyperparameters that minimized the validation loss over 10 epochs was chosen. Hyperparameters were tuned using *D. melanogaster* data and used for models in all three species. The chosen model hyperparameters are given in Additional file 4. For both hyperparameter tuning and model training early stopping was used: if the loss did not decrease for 11 or more epochs, training was stopped and the model state (weights) corresponding to the lowest loss was returned. Models were trained using only training data from their respective species. Model performance was evaluated by measuring AUROCs and AUPRs using test data from their respective species.

### Multi-context detection of homology

For the comparative genomics analyses, we used a custom reference-agnostic multi-context approach to detect homologous regions. In brief, we used the reference-free multiple sequence aligner Cactus [64], its associated tools, and custom scripts to map sets of ATAC-seq peaks from their native *Drosophila* species to both other species. We then compared the sets of peaks in the context of every species' genome to determine if they were in the same genomic location and thus presumptively orthologous. We defined conserved peaks as those with an ortholog from another species, and we defined species-specific peaks as those with no ortholog from either species.

#### Multiple sequence alignment

To compare the orthologous peaks across species, we used a multiple sequence alignment of seven related *Drosophila* species. We generated the alignment using Cactus v. 1.0.0 [64], using a phylogenetic guide tree from timetree.org [65].

#### Peak mapping

Peaks identified within a given species were mapped to each of the other studied *Drosophila* species by using the above MSA with halLiftover v2.1 (packaged with Cactus; [66]). Since halLiftover produces very many intervals for each input interval (ranging from $2.5 \times$ to $20 \times$), we used custom scripts to combine mapped intervals originating from the same peak that were within 100 bp of each other (nearest end to nearest end) into a single interval spanning the union of those intervals. We additionally only retained those intervals that reciprocally re-mapped with identical processing to within 100 bp of the original peak.

#### Detection of orthologous peaks

To determine whether peaks were conserved or not, we assumed that peaks in the same orthologous genomic location originating from different species were presumptively conserved homologs. So, we compared the set of peaks detected in each species against the set of peaks detected in every other species in the context of every species' genome (that is, *D. melanogaster* peaks vs. *D. simulans* peaks' orthologous locations in *D. melanogaster*; *D. melanogaster* peaks' orthologous locations vs. *D. simulans* peaks in *D. simulans*; *D. melanogaster* peaks' orthologous locations vs. *D. simulans* peaks' orthologous locations in *D. yakuba*, etc.). We called peaks within 100 bp of another peak in any species context as associated and presumptively homologous. Conversely, we called a peak that never had another peak within 100 bp in any species context as species-specific. To avoid ambiguity, we considered only those peaks with at least one orthologous location in the other two species. For analyses using "species-specific" peaks, we additionally excluded those peaks that had more than one orthologous location in the other two species, though we used such peaks for determining the presence or absence of homologs.

### Cross-species model performance

AUROCs and AUPRs were calculated for the *D. melanogaster* model on the *D. simulans* test set. Subsequently, these were respectively divided by the AUROCs and AUPRs of the *D. simulans* model on the *D. simulans* test set. Cross-species performance was similarly calculated for the other species combinations.

### *Aedes aegypti model performance*

Fastq files were obtained from NCBI BioProject PRJNA418406 [37]. The files were processed and peaks called as done for our *Drosophila* data.

### Saturation in silico mutagenesis

Model outputs for every possible single base-pair substitution were calculated for each example in the test set. Importance scores for each base-pair were calculated as follows. Let the raw model outputs be denoted $p_{ref}$ for the reference base and $p_1$, $p_2$, and $p_3$ for the three alternative (mutated) bases. The importance score for the base was calculated using the following equation:

$$\text{logit}(p_{ref}) - \frac{\text{logit}(p_{ref}) + \text{logit}(p_1) + \text{logit}(p_2) + \text{logit}(p_3)}{4}.$$

where the logit function is defined as $\text{logit}(p) = \log\frac{p}{1-p}$. Since the raw model outputs can be interpreted as the probability that a given input contains a peak, importance scores are effectively a log-odds ratio of there being a peak with the reference base versus there being a peak with an "average" base.

### Identification of motifs predictive of chromatin accessibility using TF-MoDISco

Importance scores from saturation in silico mutagenesis were used as inputs into TF-MoDISco [39]. To determine if candidate motifs identified by TF-MoDISco reflected known transcription factor binding motifs, Tomtom [67] was used to search candidate motifs against the JASPAR core insects database. Hits were considered significant at an FDR < 0.05.

### *Strong-selection weak-mutation experiments*

For SSWM experiments, sets of 1000 peaks (in both tissues) or non-peaks (in both tissues) were randomly chosen from chromosome arm 3L. A non-peak example would be evolved for increased accessibility in head tissue as follows. Every substitution a single nucleotide away would be made (3000 in total), and the model's output would be calculated. The mutation that increased the accessibility the most was retained for the next generation. Specifically, we chose the mutation that increased the log odds of accessibility (the logit of the model output) the most. We evolved each example under SSWM for 30 generations. In the case of opposing selection in the two tissues, the mutation that maximized the sum of the log odds (in the increasing tissue) and $-1$ times the log odds (in the decreasing tissue) was chosen. Similarly, if in silico selection was used to increase accessibility in both tissues simultaneously, the sum of log odds was maximized in each generation. A similar logic was used to decrease accessibility of peak examples.

## Supplementary Information

---

**Additional file 1.** Sequencing library statistics.

**Additional file 2: Fig S1.** Summary of peak properties. **Fig S2.** Performance of models on other test chromosomes. **Fig S3.** GC content is very weakly predictive of chromatin accessibility. **Fig S4.** Considering only non-peak examples that are proximal to peak examples does not qualitatively impair model performance. **Fig S5.** Conservation of peaks across species based on tissue. **Fig S6.** Cross-species performance of models – AUPR ratios. **Fig S7.** Genomic divergence of peaks. **Fig S8.** Species-specific losses also show model outputs between peaks and non-peaks. **Fig S9.** Low coverage sequences orthologous to species-specific peaks show model outputs between peaks and non-peaks. **Fig S10.** The relative change in model outputs vs. relative change in sequence coverage. **Fig S11.** Effect sizes of individual mutations in in silico mutagenesis. **Fig S12.** AUPRs for sliding window mutagenesis. **Fig S13.** Predicted chromatin accessibility changes as a result of deep mutation. **Fig S14.** Strong-selection weak-mutation experiments on accessible sequences.

**Additional file 3.** Motifs predictive of chromatin accessibility.

**Additional file 4.** Neural network model hyperparameters.

**Additional file 5.** Review history.

---

### Acknowledgements
We thank Andrew Kern at the University of Oregon, Viviana Risca at The Rockefeller University, and Christina Leslie, Vianne Gao, and Alli Pine at the Sloan-Kettering Institute for critically reading the manuscript. We also thank the members of the Zhao Laboratory for the helpful discussions.

### Peer review information

### Review history
The review history is available as Additional file 5.

### Authors' contributions
S.K., E.B.Z., and L.Z. conceived the study. N.S., S.M.D., and S.B. generated the data. S.K. and E.B.Z. performed all the analyses, with A.G. helping with the comparative genomics of ATAC-seq peaks and X.W. assisting in testing the model performance. S.K., E.B.Z., and L.Z. wrote the manuscript.

### Funding
This work was supported by the National Institutes of Health (NIH) MIRA R35GM133780, the Robertson Foundation, a Rita Allen Foundation Scholar Program, a Vallee Scholar Program (VS-2020-35), an Allen Distinguished Investigator Award from the Paul G. Allen Family Foundation, and a Monique Weill-Caulier Career Scientist Award to L.Z. E.B.Z. was supported by a Medical Scientist Training Program grant from the National Institute of General Medical Sciences of the National Institutes of Health under award number T32GM007739 to the Weill Cornell/Rockefeller/Sloan Kettering Tri-Institutional MD-PhD Program. A.G. was supported by NIH NRSA T32 training grant T32GM066699 and the Rosemary Grant Award from the Society for the Study of Evolution. The content of this study is solely the responsibility of the authors and does not necessarily represent the official views of the funders.

### Availability of data and materials
Our ATAC-seq data are deposited under NCBI bioProject accession number PRJNA837806 [68]. *Aedes aegypti* Fastq files were obtained from NCBI BioProject PRJNA418406 [37]. Relevant scripts and intermediate files can be found in our GitHub repository: https://github.com/LiZhaoLab/DL_ATAC/ [69] and https://doi.org/10.5281/zenodo.8381363 [70].

---

## Declarations

### Ethics approval and consent to participate
Not applicable.

### Competing interests
The authors declare that they have no competing interests.

---

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
