## [**Additional file 5.** Review history. · Genome Biology]

Review History

First round of review

Reviewer 1

Are you able to assess all statistics in the manuscript, including the appropriateness of statistical tests used? Yes, and I have assessed the statistics in my report.

Comments to author:

In this study, Khodursky and co-authors used convoluted neural network to train models that predict ATAC-seq peaks from DNA sequences. They used ATAC-seq data from the head and testis of three closely related *Drosophila* species and trained models that reasonably predict ATAC-seq peaks in distantly related *Drosophila* and even mosquito species. Intriguingly, while they found that conserved and species-specific peaks have similar between-species sequence divergence, they reported that, compared to species-specific peaks have better predictive model output than their counterparts in non-peak species. The authors interpreted that these ATAC-seq peaks were poised for evolution. Finally, the authors performed in silico mutation experiment and concluded that the sequence determinant of open chromatin regions requires small motifs and is robust to most mutational changes.

I have some comments that hope the authors could address to improve the robustness of the species-specific vs conserved peaks comparison and the in silico mutational experiment. I also hope the authors could provide more biological "interpretation" for the major advance that their study may bring to the field.

Technical comments - I hope the authors will address these comments by revising data analyses and/or providing additional technical clarifications.

1. While the in silico mutational assay provides some insights, these conclusions need to be validated by empirical data. Authors could use ATAC-seq data in similar tissues of other wildtype *D. melanogaster* strains, which I believe have been generated by either the authors' lab or other labs. Most of these strains should have been full genome sequenced, which should allow assaying the impacts of SNPs and insertion/deletions.
2. The precision/recall curve for testis does not deviate much from the diagonal line, which suggests that the model does not have strong predictive power. It would be helpful if the authors could address/discuss this result. In addition, the performance for the head seems to be better than for the testis. Any biological reasons for such a difference?
3. Results in Figure S6, S7, and Figure 3 are very interesting, but somehow do not seem to reconcile with each other. Figure S6 suggests that the sequences are not changing more dramatically for species-specific peaks than for conserved peaks, but Figure 3 and Figure S7 suggest that the model output, which presumably depends on sequence, does differ between these two types of peaks.

For Figure S6, the authors found that the between-species divergences for conserved and species-specific peaks are similar. But how about the divergence between the two non-peak species vs between one peak-one non-peak species? For instance, in Figure A, it will be helpful to compare

Dmel vs Dsim/Dyak and Dsim vs Dyak for conserved and species-specific peaks separately (maybe in X-Y plot). In order to support authors' current interpretation, such comparisons need to be similar for conserved and species-specific peaks.

In Figure S7, the authors did not find a similar trend as in Figure 3 that specie-specific loss has lower model output than those in the other species. (Only *D. simulans* head result is significant). This weakens the support for the authors' conclusion (the non-peaks are poised to evolve into peaks). It may be informative to show the other two species (which are now lumped together in the purple box) separately. This can provide two independent reference points for the species-specific peaks/loss and may reveal an overall evolutionary trend. For instance, for mel-specific peaks, whether the model output for *D. simulans* and *D. yakuba* is similar (then supporting the authors' conclusion) or *D. simulans* is more similar to *D. melanogaster* peak (then suggesting that the signals may merely be due to the expected sequence divergence between species).

A related question for Figure 3 and Figure S7 - are all MWU tests paired, which should be the more appropriate test? Showing such data in X-Y plot (for both species-specific peaks and conserved peaks involving the same species) should also help the interpretation.

4. Questions about the training data set

(1) I am curious why the authors chose 3L peaks as test data and peaks on other chromosomes as training data. Providing some biological/technical reasons would be helpful. In addition, it would be helpful to demonstrate that models trained on 3L (plus several other chr) perform equally well.

(2) It is unclear results from which training datasets are presented in the study. The dataset with matching non-peak (500bp flank) has better data balance and would be more appropriate.

(3) It is unclear how the authors address the issue with GC content in testing the predictive power of their model. Did they use non-peak with matched GC content? If so, this may also represent a better training dataset and should be used for the following analysis.

5. In Figure 4D, phyloP, which captures evolutionary changes over long evolutionary time-scale, may not be as informative as using divergence between the studied species.

Interpretation comments - I mainly hope the authors include some discussions on the following comments. No additional analyses are needed.

1. ATAC-seq peaks only inform that the chromatin is open; they do not provide additional information about the types of underlying sequence (promoter vs enhancer; which types of binding proteins). Towards the end of the study, the authors still ended up identifying motifs and referenced that to known transcription factor binding motifs. In this sense, it is unclear whether the ability to know that a chromatin region is open without performing ATAC-seq experiment may help address various biological interests. Providing some examples/interpretations will help readers appreciate the importance of the study.

2. The findings that the sequence determinant of open chromatin is robust against mutations and that there is no sequence divergence difference between conserved vs non-conserved peaks (but see above) may suggest that trans factors play an important or even a dominant role in determining chromatin state. If this is true, what would be the value/importance of predicting cis elements in the evolution study the authors referred to?

Reviewer 2

Are you able to assess all statistics in the manuscript, including the appropriateness of statistical tests used? Yes, and I have assessed the statistics in my report.

Comments to author:

This paper develops a hybrid convolution-attention neural network to predict ATAC-seq peaks using only local DNA sequences as input. The data are from head and testis of *D. melanogaster*, and

The accuracy of the predictions of chromatin openness at first seem exceptionally good, until you realize that maybe the primary thing that the model is doing is predicting whether or not a region of the genome is constitutive heterochromatin (i.e. densely compacted, with polycomb repression, etc.). A great many papers have been written about the determinants of this kind of heterochromatin, and we know many motifs responsible for the establishment and maintenance of it. To my knowledge, however, this may be the first ML approach that integrates all of that information and builds a comprehensive prediction model. This is definitely a worthwhile thing. On the other hand, the paper is written as though the model is an overall predictor of openness of chromatin. The authors also make claims about a remarkable degree of conservation of sequence specification of chromatin state, and it is important to know whether the primary thing being predicted is a yes/no call on whether a region is constitutive heterochromatin.

The complete lack of citation of the ENCODE project is a bit strange. I understand that the goal is to predict ATAC-seq output, but when there exist much more detailed experimental results on more subtle differences in chromatin state, it would seem worth citing that work and contrasting it to the present goal. Several of the ENCODE papers touch on approaches to model and predict chromatin states.

The comparison to ChromHMM (Ernst and Kellis 2012 PMID: 22373907; Ernst and Kellis 2017 PMID: 29120462) may be instructive. This tool takes a series of chIP-seq data and builds an HMM for predicting 18 distinct chromatin states. (Note that many other similar models have been devised and published, and are reviewed in Ernst and Kellis 2017). One of these states is the kind of constitutive heterochromatin described above. The present paper relies not on chIP-seq data, but instead attempts to make predictions from DNA sequence alone. It is possible that the current model basically only distinguishes between state 0 and all other states are lumped together. In this case, a more accurate description of the model is that it is a predictor of

constitutive heterochromatin. This may be why it is successful in predicting chromatin states in species other than the ones the model is tuned to - constitutive heterochromatin has evolutionarily conserved signals that recruit polycomb. Those other (euchromatic) regions are likely to be where the action is, in terms of tuning of chromatin state to achieve appropriate gene expression levels. These regions display tissue-specific and developmental-time specific changes in chromatin state (Kharchenko et al. 2011 PMID: 21179089). These are also the regions where one expects variation to result in relevant changes in gene expression. GWAS hits that map all over the genome, but which tend to be overwhelmingly noncoding are also not within constitutive heterochromatin. One does not picture much adaptive evolution occurring at genomic regions that are constitutive heterochromatin.

To tell whether the above is an accurate assessment will require a bit of work. The model needs to be trained on regions of the genome after excluding constitutive heterochromatin, or, alternatively, on focused genomic regions where the chromatin state is known to be variable across tissues or development. One would guess that this would be a far tougher challenge, with much lower prediction accuracy and much poorer transferability of a model to other species. Hopefully I am wrong, and the model retains reasonable accuracy even after masking heterochromatin, or focusing only on variable-state regions.

A far more ambitious (future!) effort would be to try to predict the 18 chromatin states defined by Ernst and Kellis 2017. There appears to be excellent data to tune such a model, and the binding site motifs of most of the factors are likewise well known. One might not be surprised to get reasonably good prediction of this fine-grained chromatin state from nothing other than the DNA sequence.

I am not a ML expert, but there are some statements that seem off to me - like on line 72 where it is claimed that once the model included attention layers, "in fact, no convolutional layers are strictly necessary." I suspect there are restriction condition only where this might be true.

Response to Reviewers' comments

Reviewer #1: In this study, Khodursky and co-authors used convoluted neural network to train models that predict ATAC-seq peaks from DNA sequences. They used ATAC-seq data from the head and testis of three closely related Drosophila species and trained models that reasonably predict ATAC-seq peaks in distantly related Drosophila and even mosquito species. Intriguingly, while they found that conserved and species-specific peaks have similar between-species sequence divergence, they reported that, compared to species-specific peaks have better predictive model output than their counterparts in non-peak species. The authors interpreted that these ATAC-seq peaks were poised for evolution. Finally, the authors performed in silico mutation experiment and concluded that the sequence determinant of open chromatin regions requires small motifs and is robust to most mutational changes.

Response: Thank you for your positive evaluation, helpful suggestions, and comments. We have now revised the manuscript according to your comments. We hope the reviewer find our response and revision adequate.

I have some comments that hope the authors could address to improve the robustness of the species-specific vs conserved peaks comparison and the in silico mutational experiment. I also hope the authors could provide more biological "interpretation" for the major advance that their study may bring to the field.

Technical comments - I hope the authors will address these comments by revising data analyses and/or providing additional technical clarifications.

1. While the in silico mutational assay provides some insights, these conclusions need to be validated by empirical data. Authors could use ATAC-seq data in similar tissues of other wildtype D. melanogaster strains, which I believe have been generated by either the authors' lab or other labs. Most of these strains should have been full genome sequenced, which should allow assaying the impacts of SNPs and insertion/deletions.

Response: Thank you for this suggestion. We indeed had previously generated ATAC seq data from the testicular tissue of 6 inbred DGRP strains. We then calculated mean CPM values within peak regions for those strains (which we had identified in our original *D. melanogaster* reference strain (2057)). Then we applied our previous saturation *in silico* mutagenesis results to predict the change in model outputs for these strains. Specifically, we computed the $\text{logit}(\text{strain prediction}) - \text{logit}(\text{reference prediction})$ for each peak in the reference strain. Even though they are likely important, we did not consider indels (since it isn't immediately clear how to modify the input sequence if the length is altered) and assumed that mutations would affect model output in an additive fashion. Despite these substantial assumptions/approximations and the fact that our model was not trained to predict accessibility CPM in peak regions – rather just the presence or absence of a peak – we observed small but positive correlations between changes in model output and changes in observed CPM for all strains (under randomness assuming independence $p=0.016$ for all correlations being positive). In 3/6 strains, we observed significant positive correlations between the change in CPM and the change in model output across peaks on 3L. Our results are shown in the tables below. Once again, we would like to emphasize that our model was not trained to predict CPM – just the absence or presence of a peak – yet we still observe statistically significant correlations in 3 of 6 strains (irrespective of magnitude of model output change) and directionally-correct correlations in the other 3. This of course includes technical variation and trans-regulatory variation that our model cannot capture, thus suggesting to us excellent performance on an out-of-context task.

DGRP Strain	Spearman R	P-value
R304	0.044	0.208
R307	0.038	0.268
R357	0.090	0.0154
R360	0.091	0.011
R399	0.060	0.091
R517	0.090	0.0085

Only considering peaks where logit model output changes by more than 0.5 (up or down)

DGRP Strain	Spearman R	P-value
R304	0.039	0.134
R307	0.024	0.350
R357	0.073	0.0076
R360	0.070	0.0085
R399	0.045	0.093
R517	0.074	0.0039

Only considering peaks where logit model output changes by more than 0.2 (up or down)

DGRP Strain	Spearman R	P-value
R304	0.031	0.146
R307	0.022	0.295
R357	0.049	0.024
R360	0.058	0.0062
R399	0.026	0.213
R517	0.054	0.0093

Only considering peaks where logit model output changes at all. Depending on strain, for approximately ~700-850 out of the 3000 peaks, model outputs remained the same due to a lack of SNP variation. Those peaks were discarded.

2. The precision/recall curve for testis does not deviate much from the diagonal line, which suggests that the model does not have strong predictive power. It would be helpful if the authors could address/discuss this result. In addition, the performance for the head seems to be better than for the testis. Any biological reasons for such a difference?

Response: We thank the reviewer for these comments. Generally speaking, the random expectation for the area under the precision-recall curve is the fraction of all examples that are positive (peaks in our case). So for our data, if the model was randomly guessing, we would expect areas under the precision-recall curve to be around 0.2 or even lower while we observe values around .65 for testis and over 0.75 for head tissue. The higher performance for head tissue could possibly be explained by several factors. First, the increased imbalance of the data for testis (relatively fewer peaks) could partially explain the lower AUPR values we found for testis (since the increased imbalance would also lower the baseline expectation). Additionally, differences between the head and testis could be due to biological factors. For example, in spermatogenesis, histones are gradually replaced by protamines. Protamines replace histones

during the elongation and condensation of spermatids in spermatogenesis and allow the DNA to be packed more densely. The possible “noise” in this transition and fewer open chromatin regions may also contribute to the differences between head and testis.

3. Results in Figure S6, S7, and Figure 3 are very interesting, but somehow do not seem to reconcile with each other. Figure S6 suggests that the sequences are not changing more dramatically for species-specific peaks than for conserved peaks, but Figure 3 and Figure S7 suggest that the model output, which presumably depends on sequence, does differ between these two types of peaks.

Response: This observation is exactly what we find exciting. We believe that while the *amount* of genetic variation is approximately similar, which *exact* variants become fixed are different between conditions. We interpret this as evidence supporting a complex sequence-dependent *cis*-regulatory mechanism governing chromatin accessibility. Additionally, we demonstrate that a handful of very specific/adaptive mutations can dramatically alter model output (new **Figure 7**) – however, most random mutations have little effect (see our robustness results). Therefore, purely examining sequence divergence (of our 1kb input) is likely not very informative when thinking about model outputs.

For Figure S6, the authors found that the between-species divergences for conserved and species-specific peaks are similar. But how about the divergence between the two non-peak species vs between one peak-one non-peak species? For instance, in Figure A, it will be helpful to compare *Dmel* vs *Dsim/Dyak* and *Dsim* vs *Dyak* for conserved and species-specific peaks separately (maybe in X-Y plot). In order to support authors' current interpretation, such comparisons need to be similar for conserved and species-specific peaks.

Response: Thanks for the suggestion of an X-Y plot; we have revised **Supp. Figure 6** (now **Supp. Figure 7**). We think this illustrates the lack of difference more clearly.

We also examined the differences between the nonpeak regions orthologous to peaks (below; both conserved or unique to the species listed at the top). Note that an X-Y comparison here is not appropriate, so we present it using the older 1-dimensional plot.

As you can see from the revised supplemental figure and the data shown above, while some peaks have greater or lesser rates of substitution, this does not correlate with differences in conservation, supporting our interpretation.

In Figure S7, the authors did not find a similar trend as in Figure 3 that specie-specific loss has lower model output than those in the other species. (Only D. simulans head result is significant). This weakens the support for the authors' conclusion (the non-peaks are poised to evolve into peaks). It may be informative to show the other two species (which are now lumped together in the purple box) separately. This can provide two independent reference points for the species-specific peaks/loss and may reveal an overall evolutionary trend. For instance, for mel-specific peaks, whether the model output for D. simulans and D. yakuba is similar (then supporting the authors' conclusion) or D. simulans is more similar to D. melanogaster peak (then suggesting that the signals may merely be due to the expected sequence divergence between species).

Response: Thank you for this comment. As suggested, we examined whether trends are more illuminating when separating out *D. yakuba* due to its greater evolutionary distance. Using a paired Wilcoxon signed-rank test between species-specific losses and orthologous peaks reveals additional statistically significant differences in *D. melanogaster* head and recapitulates the result in *D. simulans* head. The current figure in our opinion kept the key information while not misleading the audience. We are not concerned overall, as there are substantially fewer of these species-specific losses (100-200) compared to species-specific peaks (400-600) on chromosome arm 3L. So our species-specific analysis may be underpowered.

A related question for Figure 3 and Figure S7 - are all MWU tests paired, which should be the more appropriate test? Showing such data in X-Y plot (for both species-specific peaks and conserved peaks involving the same species) should also help the interpretation.

Response: The Mann-Whitney U (MWU) tests are not paired, as an appropriate pairing does not exist due to the lack of a 1:1 relationship. For instance, the proposed comparison of species-specific peaks and conserved peaks within the same species is not feasible in X-Y space (as we demonstrate in Figure 2A, where the vast majority of peaks are conserved). Consequently, a peak in, say, *D. melanogaster* is either species-specific or conserved. Infrequently, it can be specific to two species, but we did not analyze these few peaks due to their insufficient numbers.

4. Questions about the training data set

(1) I am curious why the authors chose 3L peaks as test data and peaks on other chromosomes as training data. Providing some biological/technical reasons would be helpful. In addition, it would be helpful to demonstrate that models trained on 3L (plus several other chr) perform equally well.

Response:

Response to (1): To aid in our comparative analysis, we were interested in using a test chromosome arm that was autosomal and syntenic/homologous across the three species (corresponding to the same Muller element). A pericentric inversion in chromosome 2 in *D. yakuba* ruled out 2L and 2R. Out of 3L and 3R, we chose 3L since it is slightly smaller than 3R – leaving slightly more data for training – although this decision was not particularly critical. To help address the reviewer's concerns about our choice of 3L, we trained models leaving one arm out for testing at a time and found that performance was similar for all major chromosome arms (shown below) (**See new Supplementary Figure 2 in text**).

(2) It is unclear results from which training datasets are presented in the study. The dataset with matching non-peak (500bp flank) has better data balance and would be more appropriate.

Response to (2): We trained our models using all of the negative data available (excluding, of course, examples from the testing chromosome). However, all of the results shown in the manuscript show performance on test data from chromosome arm 3L. We were indeed concerned about our choice of negative examples; however, when we found that model performance remained excellent in discriminating peaks and non-peaks even when only considering non-peaks that flanked peaks (as shown in the manuscript, Supp. Fig. 4), we decided to retain our model that was trained on all possible non-peaks.

(3) It is unclear how the authors address the issue with GC content in testing the predictive power of their model. Did they use non-peak with matched GC content? If so, this may also represent a better training dataset and should be used for the following analysis.

Response to (3): Similarly, for point (3), when we found that GC content was very weakly predictive (if at all) of chromatin accessibility in *Drosophila*, we decided to continue with our original model without GC matching. We believe that changing the balance of our data set (by only using flanking regions for example) would have limited impact on our conclusions since model performance remained very similar.

5. In Figure 4D, phyloP, which captures evolutionary changes over long evolutionary time-scale, may not be as informative as using divergence between the studied species.

Response: We thank the reviewer and certainly agree that phyloP is an imperfect metric to use. However, we specifically use phyloP as it provides a base-pair level conservation statistic – which can then be correlated with our base-pair level in silico mutagenesis results. Simple divergence would be a binary measure at a base-pair level; estimating total substitutions at a given site would, in our view, add significant complexity for unclear benefit.

Interpretation comments - I mainly hope the authors include some discussions on the following comments. No additional analyses are needed.

1. ATAC-seq peaks only inform that the chromatin is open; they do not provide additional information about the types of underlying sequence (promoter vs enhancer; which types of binding proteins). Towards the end of the study, the authors still ended up identifying motifs and referenced that to known transcription factor binding motifs. In this sense, it is unclear whether the ability to know that a chromatin region is open without performing ATAC-seq experiment may help address various biological interests. Providing some examples/interpretations will help readers appreciate the importance of the study.

Response: Besides identifying known motifs, we also identified 90 motifs that could not be significantly aligned to known motifs in both the general JASPAR and JASPAR insect databases (FDR>0.05) (**Supplementary File 2**). We have added logo plots for two of these motifs to **Figure 8** (previously **Figure 7**). The strength (and the main focus) of our study is in utilizing the models to answer evolutionary questions related to conservation, innovation, and robustness. We found that the basis of regulation is conserved, robust, and possibly redundant, and that species-specific peaks arise in regions of the genome poised for accessibility. We do think the motif analysis is important for biologists, since little is known about tissue-specific binding motifs in *Drosophila*.

2. The findings that the sequence determinant of open chromatin is robust against mutations and that there is no sequence divergence difference between conserved vs non-conserved peaks (but see above) may suggest that trans factors play an important or even a dominant role in determining chromatin state. If this is true, what would be the value/importance of predicting cis elements in the evolution study the authors referred to?

Response: Although it is certainly possible that trans factors contribute significantly to chromatin accessibility, we believe that local/cis sequence is also very informative by itself, hence the high AUCs for our models (which only consider local sequence). However, it is possible that evolution in trans factors or more distal sequence elements could possibly be of particular importance in the emergence of species-specific peaks – hence the lower model outputs for species-specific peaks than for conserved peaks, and we have added some text to the discussion on this (see **lines 494-496**). Even so, the *cis* sequence still is informative/important for novel peaks as the model outputs are significantly higher for those sequences than for non-peak sequences. Additionally, the similarity in sequence divergence between conserved and novel peaks is entirely consistent with our observation of a small (rather than

large) difference in model output between novel peak sequences and their orthologous non-peak sequences.

Another factor that might be of some relevance when discussing sequence divergence is that 4-5 very carefully placed mutations can turn a non-peak into a peak (see new **Figure 7**). A handful of mutations would do little to alter the average sequence divergence of a 1kb sequence. In other words, it seems plausible that much of the observed sequence divergence contributes little to accessibility changes (which is also in line with our random mutagenesis/robustness results), while only a few of the mutations contribute substantially – and their signal might be largely concealed by mutations which do little to alter accessibility.

Reviewer #2:

*This paper develops a hybrid convolution-attention neural network to predict ATAC-seq peaks using only local DNA sequences as input. The data are from head and testis of *D. melanogaster*, and the accuracy of the predictions of chromatin openness at first seem exceptionally good, until you realize that maybe the primary thing that the model is doing is predicting whether or not a region of the genome is constitutive heterochromatin (i.e. densely compacted, with polycomb repression, etc.). A great many papers have been written about the determinants of this kind of heterochromatin, and we know many motifs responsible for the establishment and maintenance of it. To my knowledge, however, this may be the first ML approach that integrates all of that information and builds a comprehensive prediction model. This is definitely a worthwhile thing.*

Response: Thank you for your positive evaluation, which we greatly appreciate. While we agree that the model performance is indeed impressive, we do not attribute this to constitutive heterochromatin. The reason is that only a small minority of the *Drosophila* reference genome consists of constitutive heterochromatin. The genome of *D. melanogaster* is approximately 180MB in size, but the reference genome only accounts for about 130MB. The remaining regions not included in the genome may be largely constitutive heterochromatin, but since they are not sequenced, they are not used in our analysis and do not affect our results. As detailed in our below response, only a small proportion of the sequenced genome consists of constitutive heterochromatin, while our non-peak examples account for the majority of the genome. However, we agree with the reviewer that if we were to study other species with a high percentage of constitutive heterochromatin, a separate model should be used to evaluate the effect of constitutive heterochromatin on model performance.

On the other hand, the paper is written as though the model is an overall predictor of openness of chromatin. The authors also make claims about a remarkable degree of conservation of sequence specification of chromatin state, and it is important to know whether the primary thing being predicted is a yes/no call on whether a region is constitutive heterochromatin.

Response: We thank the reviewer for this comment and agree that this is a valid concern. Recent sources have suggested that up to 30% of the *D. melanogaster* genome could be constitutive heterochromatin (Marsano et al., 2019, PMID: 31320181). However, much of that heterochromatin is not included in the Release 6 version of the reference genome (regions not included in the reference genome are not relevant to our analysis, as they are not included in either training, validation, or testing), which is the reference that our work based on. Marsano et al. estimate that ~16-17 Mb of the R6 reference is constitutive heterochromatin, excluding the Y chromosome (which we exclude in our own analysis). In contrast, our dataset (training + validation + testing) consists of over 100 Mb of nonpeak examples (103 Mb for head and 108 Mb for testis). Even in the hypothetical edge case of all the constitutive heterochromatin being included in the reference genome and accounting for 30% of the reference (doubling Marsano *et al.*'s estimate), it would still comprise a minority of our nonpeak examples which amount to approximately 80% of the total sequence of the major chromosomes. Therefore, it is implausible that our model simply

discriminates between constitutive heterochromatin and euchromatin (and facultative heterochromatin). We have added this to our discussion section (**lines 464-473**).

The complete lack of citation of the ENCODE project is a bit strange. I understand that the goal is to predict ATAC-seq output, but when there exist much more detailed experimental results on more subtle differences in chromatin state, it would seem worth citing that work and contrasting it to the present goal. Several of the ENCODE papers touch on approaches to model and predict chromatin states.

Response: We have included references for several ENCODE publications and have included a brief section in the discussion on how a model that distinguishes between more specific chromatin states could be informative (**lines 547-552**).

The comparison to ChromHMM (Ernst and Kellis 2012 PMID: 22373907; Ernst and Kellis 2017 PMID: 29120462) may be instructive. This tool takes a series of chIP-seq data and builds an HMM for predicting 18 distinct chromatin states. (Note that many other similar models have been devised and published, and are reviewed in Ernst and Kellis 2017). One of these states is the kind of constitutive heterochromatin described above. The present paper relies not on chIP-seq data, but instead attempts to make predictions from DNA sequence alone. It is possible that the current model basically only distinguishes between state 0 and all other states are lumped together. In this case, a more accurate description of the model is that it is a predictor of constitutive heterochromatin. This may be why it is successful in predicting chromatin states in species other than the ones the model is tuned to - constitutive heterochromatin has evolutionarily conserved signals that recruit polycomb. Those other (euchromatic) regions are likely to be where the action is, in terms of tuning of chromatin state to achieve appropriate gene expression levels. These regions display tissue-specific and developmental-time specific changes in chromatin state (Kharchenko et al. 2011 PMID: 21179089). These are also the regions where one expects variation to result in relevant changes in gene expression. GWAS hits that map all over the genome, but which tend to be overwhelmingly noncoding are also not within constitutive heterochromatin. One does not picture much adaptive evolution occurring at genomic regions that are constitutive heterochromatin.

Response: Thank you for the comment. As discussed above, we believe that only a small fraction of our non-peak examples corresponds to constitutive heterochromatin. Even though a model that distinguishes between constitutive heterochromatin, facultative heterochromatin, euchromatin, and other chromatin states like enhancers/promoters would likely be exciting and scientifically illuminating, we believe that such a model is beyond the scope of this project and would be suitable for a follow up given appropriate genomic data. We added a paragraph in the discussion to reflect this idea (**line 547-552**). We also added relevant references suggested by the reviewer.

To tell whether the above is an accurate assessment will require a bit of work. The model needs to be trained on regions of the genome after excluding constitutive heterochromatin, or, alternatively, on focused genomic regions where the chromatin state is known to be variable across tissues or development. One would guess that this would be a far tougher challenge, with much lower prediction accuracy and much poorer transferability of a model to other species. Hopefully I am wrong, and the model retains reasonable accuracy even after masking heterochromatin, or focusing only on variable-state regions.

Response: We thank the reviewer for this comment. As discussed above, the *Drosophila* reference genome contains a relatively low fraction of constitutive heterochromatin and that would comprise a minority of our nonpeak examples. Therefore, it is very unlikely that our model discriminates between constitutive heterochromatin and euchromatin (and facultative heterochromatin). However, we agree with the reviewer that in the future, when we obtain additional epigenomic data we could fine-tune the model by studying open and closed chromatin in multiple states.

A far more ambitious (future!) effort would be to try to predict the 18 chromatin states defined by Ernst and Kellis 2017. There appears to be excellent data to tune such a model, and the binding site motifs of most of the factors are likewise well known. One might not be surprised to get reasonably good prediction of this fine-grained chromatin state from nothing other than the DNA sequence.

Response: We certainly agree that this is an interesting future direction and have included it in **lines 547-552** of our discussion.

I am not a ML expert, but there are some statements that seem off to me - like on line 72 where it is claimed that once the model included attention layers, "in fact, no convolutional layers are strictly necessary." I suspect there are restriction condition only where this might be true.

Response: Thank you for the suggestion, edited.

Second round of review

Reviewer 2

Many of my previous comments centered on the issue of the binary prediction of ATAC-seq peaks being a bit trivial if constitutive heterochromatin was in the mix. I had not realized that the reference genome used did not include this segment of the genome, and this makes a big difference. Generally I am satisfied with the paper, but there are two few issues that still could use attention.

1. One is a bit worried that the level of prediction is at a level that could miss interesting differences associated with evolutionary divergence, polymorphisms related to disease risk, etc. If such changes are in a sense "below the radar" of prediction by this method, then are the predictions really that useful?

2. There have been published methods that had a similar objective (prediction of ATAC-seq peaks from DNA sequence. These include:

a. SemanticCAP by Zhang et al. (2022 Genes) <https://pubmed.ncbi.nlm.nih.gov/35456374/>

b. Deopen by Liu et al. (2018 Bioinformatics

<https://www.ncbi.nlm.nih.gov/pmc/articles/PMC6192215/>

c. A method by Zhou et al. (2019 <https://pubmed.ncbi.nlm.nih.gov/31428792/>

d. And finally, the paper entitled, "Identification of determinants of differential chromatin accessibility through a massively parallel genome-integrated reporter assay" seems particularly relevant to the issue of predicting motifs that designate open chromatin. This paper was by Hammelman et al. (2020 Genome Research) <https://genome.cshlp.org/content/30/10/1468.full>

Response to Reviewer 2's Comments

1. One is a bit worried that the level of prediction is at a level that could miss interesting differences associated with evolutionary divergence, polymorphisms related to disease risk, etc. If such changes are in a sense "below the radar" of prediction by this method, then are the predictions really that useful?

Response:

We thank the reviewer for their comments. We believe that our models are useful in studying differences associated with evolutionary divergence, among other things. For example, in Fig 3 and associated work, we show that these trained models perform statistically significantly (managing to show different outputs for orthologous nonpeak and peak sequences), suggesting that this method is capable of capturing these effects. Second, we could theoretically consider within-species differences with a similar model; however, as described in our previous response, these strain-specific analyses are not something that our model was specifically built for (See response to Reviewer 1 comment #1 from the first revision). Models with similar architecture, however, have been applied to the study of polymorphism in a population (Zhou et al., 2016 PMID: 26301843 ; Vaishnav et al., 2022 PMID: 35264797). The strength and novelty of our paper lie in addressing evolutionary questions concerning chromatin accessibility over longer timescales. Beyond that, we also demonstrate the utility of our model in helping to answer biological questions (such as identification of motifs).

2. There have been published methods that had a similar objective (prediction of ATAC-seq peaks from DNA sequence. These include:

a. SemanticCAP by Zhang et al. (2022 Genes) <https://pubmed.ncbi.nlm.nih.gov/35456374/>

b. Deopen by Liu et al. (2018 Bioinformatics
<https://www.ncbi.nlm.nih.gov/pmc/articles/PMC6192215/>

c. A method by Zhou et al. (2019 <https://pubmed.ncbi.nlm.nih.gov/31428792/>

d. And finally, the paper entitled, "Identification of determinants of differential chromatin accessibility through a massively parallel genome-integrated reporter assay" seems particularly relevant to the issue of predicting motifs that designate open chromatin. This paper was by Hammelman et al. (2020 Genome Research)

Response:

We thank the reviewer for pointing out these references and we have added them to the introduction of the manuscript.